# Mapping effector genes at lupus GWAS loci using promoter Capture-C in follicular helper T cells

Chun Su[1,9], Matthew E. Johnson[1,9], Annabel Torres[2,9], Rajan M. Thomas[2,9], Elisabetta Manduchi [1,3], Prabhat Sharma [2], Parul Mehra[2], Carole Le Coz[4], Michelle E. Leonard[1], Sumei Lu[1], Kenyaita M. Hodge[1], Alessandra Chesi[1], James Pippin [1], Neil Romberg [4,5], Struan F. A. Grant [1,5,6,7,10] & Andrew D. Wells [2,8,10 ✉]

Systemic lupus erythematosus (SLE) is mediated by autoreactive antibodies that damage multiple tissues. Genome-wide association studies (GWAS) link >60 loci with SLE risk, but the causal variants and effector genes are largely unknown. We generated high-resolution spatial maps of SLE variant accessibility and gene connectivity in human follicular helper T cells (TFH), a cell type required for anti-nuclear antibodies characteristic of SLE. Of the ~400 potential regulatory variants identified, 90% exhibit spatial proximity to genes distant in the 1D genome sequence, including variants that loop to regulate the canonical TFH genes *BCL6* and *CXCR5* as confirmed by genome editing. SLE 'variant-to-gene' maps also implicate genes with no known role in TFH/SLE disease biology, including the kinases HIPK1 and MINK1. Targeting these kinases in TFH inhibits production of IL-21, a cytokine crucial for class-switched B cell antibodies. These studies offer mechanistic insight into the SLE-associated regulatory architecture of the human genome.

[1] Division of Human Genetics, The Children's Hospital of Philadelphia, 3615 Civic Center Boulevard, Philadelphia, PA, USA. [2] Department of Pathology, The Children's Hospital of Philadelphia, 3615 Civic Center Boulevard, Philadelphia, PA, USA. [3] Institute for Biomedical Informatics, University of Pennsylvania, 3700 Hamilton Walk, Philadelphia, PA, USA. [4] Division of Allergy and Immunology, The Children's Hospital of Philadelphia, 3615 Civic Center Boulevard, Philadelphia, PA, USA. [5] Department of Pediatrics, Perelman School of Medicine, University of Pennsylvania, 3615 Civic Center Boulevard, Philadelphia, PA, USA. [6] Division of Diabetes and Endocrinology, The Children's Hospital of Philadelphia, 3615 Civic Center Boulevard, Philadelphia, PA, USA. [7] Department of Genetics, Perelman School of Medicine, University of Pennsylvania, 3615 Civic Center Boulevard, Philadelphia, PA, USA. [8] Department of Pathology and Laboratory Medicine, Perelman School of Medicine, University of Pennsylvania, 3615 Civic Center Boulevard, Philadelphia, PA, USA. [9] These authors contributed equally: Chun Su, Matthew E. Johnson, Annabel Torres, Rajan M. Thomas. [10] These authors jointly supervised this work: Struan F. A. Grant, Andrew D. Wells. ✉email: adwells@pennmedicine.upenn.edu

G enome-wide association studies (GWAS) is an important tool in understanding the genetic basis of complex, heritable diseases and traits. However, GWAS is typically powered to identify large blocks of the genome containing hundreds of single nucleotide polymorphisms (SNP) in linkage disequilibrium (LD), any one of which could be responsible for the association of the entire locus with disease susceptibility. Moreover, ~90% of GWAS-implicated SNPs are non-coding, and per se do not identify the culprit genes. Examples of this are the *FTO* signal in obesity[1,2], and the *TCF7L2* signal in type 2 diabetes[3], where the suspected causal variant resides in an intron of the local gene, but instead regulates expression of the distant genes.

Systemic lupus erythematosus (SLE) is a complex inflammatory disease mediated by autoreactive antibodies that damage multiple tissues in children and adults[4]. An inflammatory leukocyte required for the development of SLE is the follicular helper T cell (TFH). TFH differentiate from naive CD4+ T cells in the lymph nodes, spleen, and tonsil, where they license B cells to produce high affinity protective or pathogenic antibodies[5,6]. Given their central role in regulation of humoral immune responses, genetic susceptibility to SLE is highly likely to manifest functionally in TFH.

GWAS has associated >60 loci with SLE susceptibility[7,8]. Given the paucity of immune cell eQTL data represented in GTEx, we mapped the open chromatin landscape of TFH from human tonsil to identify potentially functional SLE variants. Here, we conduct a genome-wide, promoter-focused Capture-C analysis of chromatin contacts at ~42,000 annotated human genes at ~270 bp resolution to map these variants to the genes they likely regulate. This approach, which we used recently to identify new effector genes at bone mineral density loci[9], only requires three samples to make valid interaction calls, and does not require material from SLE patients or genotyped individuals. By design, this approach does not determine the effect of variants in the system, but rather, uses reported variants as 'signposts' to identify potential gene enhancers in normal tissue. We show that most SLE-associated variants do not interact with the nearest promoter, but instead connect to distant genes, many of which have known roles in TFH and SLE. Using CRISPR/CAS9 genome editing, we validate several of these SLE-associated regions, revealing a requisite role in regulating their connected genes. Finally, we experimentally verify roles for two kinases implicated by this variant-to-gene mapping approach in TFH differentiation and function, identifying potential drug targets for SLE and other antibody-mediated diseases.

## Results

**Human tonsillar naive T cell and TFH open chromatin landscapes.** The vast majority (>90%) of the human genome is packed tightly into cellular chromatin and is not accessible to the nuclear machinery that regulates gene expression[10]. Consequently, >95% of transcription factor and RNA polymerase occupancy is concentrated at regions of open chromatin[10], and thus the map of accessible chromatin in a cell essentially defines its potential gene regulatory landscape. As a step toward defining the disease-associated regulatory architecture of SLE, we focused on human TFH cells, which are required for the production of pathogenic antibodies by autoreactive B cells[4]. Tonsillar TFH are derived from naive CD4+ T cell precursors, and represent a population of cells in healthy subjects that are 'caught in the act' of helping B cells to produce high-affinity, class-switched antibodies. We sorted naive CD4+CD45RO− T cells and differentiated CD4+CD45RO+CD25−CXCR5[hi]PD1[hi] TFH[11] from human tonsil and generated open chromatin maps of both cell types from three

donors using ATAC-seq[12]. A peak calling approach identified 91,222 open chromatin regions (OCR); 75,268 OCR in naive cells and 74,627 OCR in TFH (Supplementary Data 1). Further quantitative analysis of the accessibility signal at each OCR revealed a similar overall degree of genomic accessibility (~1.4%) in both cell types (Supplementary Fig. 1), however, differentiation of naive cells into TFH is associated with remodeling of 22% of the T cell open chromatin landscape, with 11,228 OCR becoming more accessible, and 8,804 becoming less accessible (Fig. 1a, Supplementary Data 1). Among all 20,032 differentially accessible regions, 20.5% (4100) reside in the promoters of genes that tended to be differentially expressed between TFH and naive cells (Fig. 1b, 1496 DEG, GSEA enrichment $p < 0.05$, absNES > 3.5, Supplementary Data 2). The functions of genes more accessible in TFH were enriched for CD28 costimulatory, G-protein, Rho GTPase, semaphorin, and TLR signaling pathways (hypergeometric test, FDR < 0.05, Supplementary Fig. 2a). Promoters less accessible in TFH are enriched for chemokine and G protein-coupled signaling pathways (Supplementary Fig. 2b). These data show that global chromatin remodeling dynamics faithfully reflect dynamic changes in gene expression during the differentiation of follicular helper T cells from their naive precursors.

**TFH open chromatin implicates causal disease variants.** Of the 2416 proxy SNPs in high LD ($r^2 > 0.8$) with sentinel SNPs currently implicated in SLE by GWAS[7,8], 8% (190) reside in 109 open chromatin regions in either naive CD4+ T cells or TFH (Supplementary Data 3). LD score regression (LDSR) analysis showed that SLE-associated genetic variation is enriched in TFH open chromatin to a greater extent than in naive open chromatin (Supplementary Table 1), further supporting the significance of TFH cells in SLE. Of these SNPs, 79% (150) are in open chromatin shared by both cell types, 6% (11) are in naive-specific OCR, and 15% (29) reside uniquely within the TFH open chromatin landscape (Fig. 2a). Relaxing the $r^2$ threshold to 0.4 uncovered additional, potentially functional SLE SNPs in open chromatin (total of 432), 91% of which (393) reside in the open chromatin landscape of TFH cells. To explore the potential significance of these open chromatin-implicated variants, we first focused on the 132 SLE proxy SNPs that reside in open promoters of protein-coding genes in TFH. Of 64 genes containing one or more open promoter variants, 62 are expressed in TFH (Supplementary Data 2).

Genes with accessible SLE variants in their promoters are expressed more highly in TFH compared to the set of all genes, or compared to a random sample of genes with open promoters in TFH. This trend is observed for SLE promoter proxies in high LD ($r^2 > 0.8$) and for proxies with relaxed LD ($r^2 > 0.4$, Fig. 2b). Eighty-three SLE proxies reside in promoters of 36 genes in the top 75% expression quantile, and 43 proxies reside in the promoters of 18 genes in the 50–75% expression quantile. Thus, 93% (123 of 132) of open promoter SLE variants are positioned at genes highly expressed in TFH. Ingenuity pathway analysis (IPA) shows these genes are enriched for transcription factors and receptors involved in SLE ($P = 4.6 \times 10^{-9}$), rheumatic disease ($P = 2.7 \times 10^{-7}$), and cell viability ($p = 6.0 \times 10^{-8}$) (Fig. 2c). This set of SLE-associated genes includes *DHCR7* and *NADSYN1*, enzymes involved in the biogenesis of vitamin D, a process known to play an important role in autoimmune disease susceptibility[13]. These results confirm that open chromatin landscapes in disease-relevant cell types represent a highly specific filter through which putatively functional variants can be distinguished from the thousands of SNP implicated by GWAS[14].

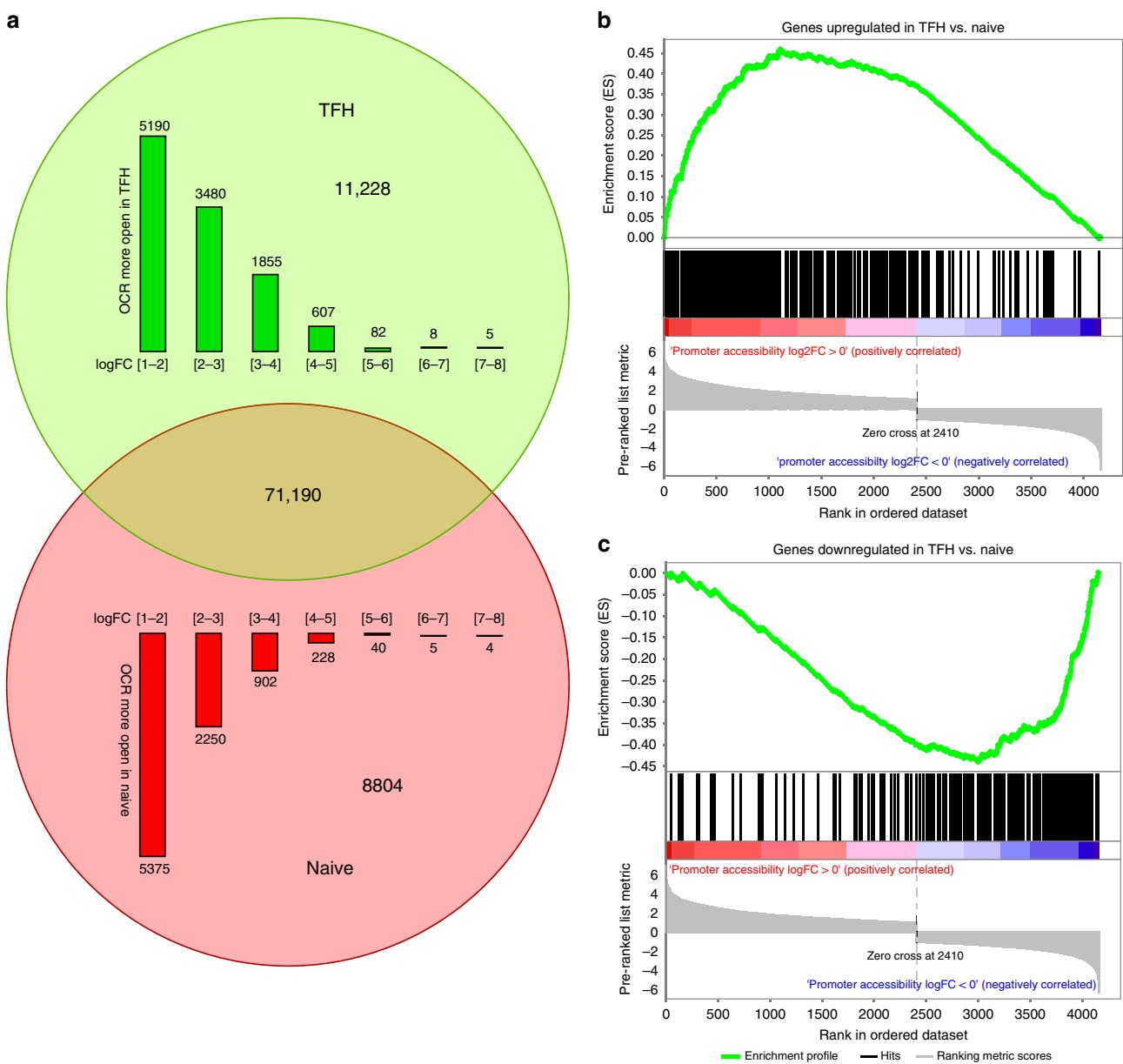

**Fig. 1 ATAC-seq analysis of open chromatin landscapes in naive and follicular helper T cells from human tonsil. a** Quantitative differences between naive and follicular helper T cell open chromatin landscapes. A total of 91,222 OCR were used as reference for differential analysis of genome accessibility. The number of statistically up- (green) or down- (red) regulated OCR in TFH compared to naive (FDR < 0.05, log2 fold change > 1 or < -1) is shown as a Venn diagram and also plotted as a function of log2 fold change. **b**, **c** GSEA enrichment analysis of genes with differential promoter accessibility at promoter regions. The log2 fold change in promoter accessibility of genes upregulated (**b**) or downregulated (**c**) in TFH vs. naive was used to generate a pre-ranked list for comparison to differentially expressed genes.

**Analysis of the three-dimensional promoter connectome structure in TFH cells.** It is relatively clear how genetic variation at a promoter might influence expression of the downstream gene. However, ~70% of the accessible SLE SNPs in TFH cells are intronic, intergenic, or otherwise not in a promoter, so how these variants regulate expression of TFH genes is not clear from 1-dimensional chromatin mapping alone. To explore the role that non-coding SLE-associated variants play in the disease-related regulatory architecture of the human genome, we derived genome-wide, three-dimensional promoter contact maps of naive and differentiated follicular helper T cells from human tonsil using promoter-focused Capture-C[9]. Our current probe set allows enrichment of 41,970 annotated promoters associated with 123,526 transcripts covering 89% of protein-coding mRNA genes

and 59% of non-coding genes in the human genome[15]. Valid hybrid reads derived from ligation of distant fragments with bait fragments were preprocessed using HiCUP[16], and significant promoter-interacting regions (PIR, score >5) were called using CHiCAGO[17]. These studies achieve an observed median resolution of 270 bp, allowing mapping of interactions between promoters and distal elements to within a span of two nucleosomes. This precision comes at the expense of power, in that fewer reads are available per fragment to call significant promoter interactions. To circumvent this problem, we called promoter interactions both at high resolution (single-fragment) and lower resolution (four-fragment) after an in silico concatenation step. Combination of both sets benefits from the precision of single DpnII fragment analysis and the power of lower resolution

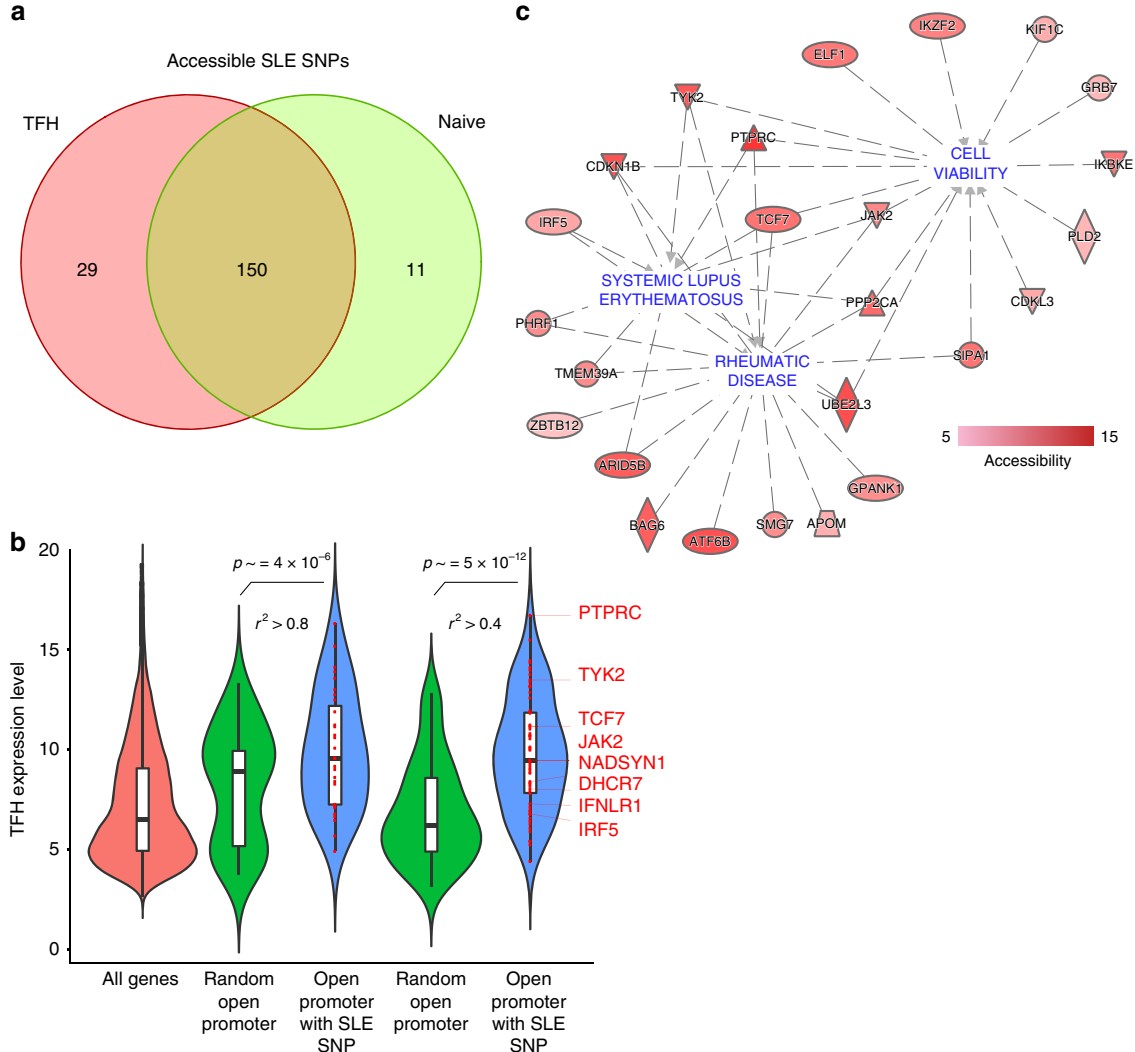

**Fig. 2 Genes harboring accessible SLE variants in naive and follicular helper T cells. a** Comparison of accessible SLE SNPs between TFH and naive tonsillar T cells. **b** Comparison of the expression levels of genes with accessible SLE SNPs in their promoters in TFH vs. all genes or a random sample of genes with no accessible SLE SNPs in their promoters. A two-side Wilcoxon rank-sum test was performed to evaluate the significance of differential expression between gene sets. **c** Ingenuity disease network for the genes with accessible SLE variants at promoters. The color gradient represents the log2 fold change in accessibility in TFH compared to naive T cells.

analyses to assemble comprehensive, 3D promoter-focused contact maps for the human 'promoterome' (Fig. 3a).

We detected a similar number of promoter interactions in both cell types (255,238 in naive cells and 224,263 in TFH), >99% of which were intra-chromosomal (*cis*). About 20% of total interactions were between two promoters, while 80% were between a promoter and an intergenic or intronic region (Supplementary Table 2). Ninety percent of promoters were found engaged in at least one interaction with another genomic region. Of these, >80% were connected to only one distal region, indicating that most promoters in these cell types exhibit very low spatial complexity. However, ~1% of all promoters exhibited significant spatial complexity, interacting with at least 4 regions, with some engaging in as many as 70 interactions. The number of connections per promoter correlated with the level of gene expression, with the most interactive promoters belonging to highly-expressed genes with known roles in TFH function (Fig. 3b). Two examples are the *IL21* and *IFNG* promoters, which are expressed and show complex connectomes in TFH but not naive cells (Fig. 3c, d, Supplementary Fig. 3a, b). Promoter-interacting regions were enriched 3-fold for open chromatin, and

2-fold for chromatin signatures associated with active transcription (H3K27ac, H3K4me1, H3K4me3—Fig. 4a and Supplementary Fig. 4a). Conversely, PIR were depleted of silencing marks (H3K27me3, H3K9me3—Fig. 4a and Supplementary Fig. 4a). These trends indicate that promoter contacts captured by this approach preferentially represent the active regulatory architectures of the associated genes.

**The promoter-open chromatin connectome of TFH cells.** To further explore the regulatory nature of spatial connections between promoters and chromatin in the nucleus, we focused on contacts between promoters and OCR, as the processes regulating transcription occur largely at accessible DNA[10]. Instead of using standard fragment-based interactions[9], we used a feature-based calling approach to generate genome-wide, open chromatin-promoter interaction landscapes in naive and follicular helper T cells (Fig. 3a). We detected 71,137 interactions between accessible promoters and open chromatin across both cell types, involving 34% of the total open chromatin landscape identified by ATAC-seq (Supplementary Table 2). We define these 31,404

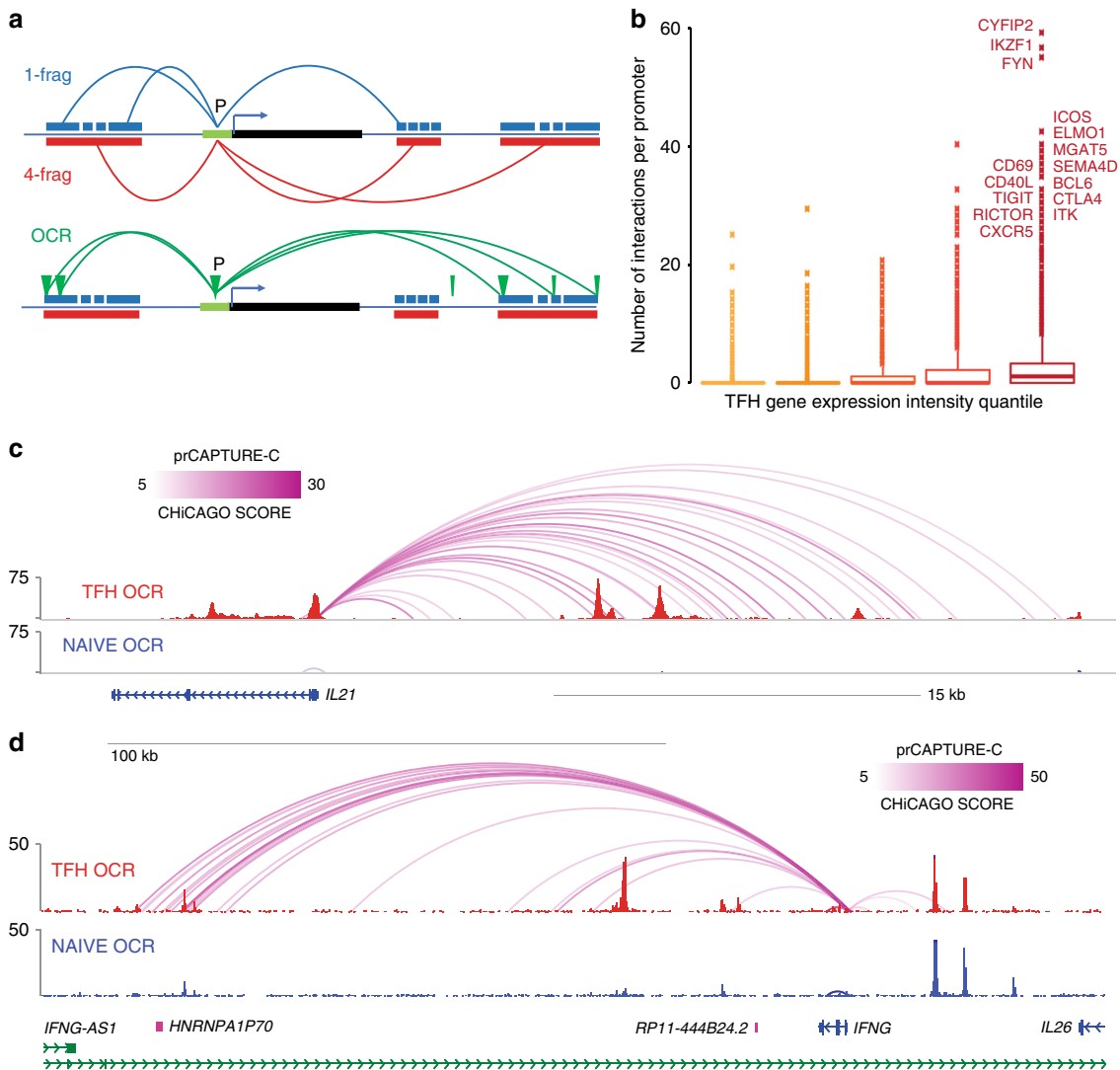

**Fig. 3 High-resolution, fragment-based Capture-C analysis of promoter connectomes in naive and follicular helper T cells. a** Cartoon depicting the approaches for 1 DpnII fragment promoter interaction analysis, 4 DpnII fragment promoter interaction analysis, and promoter-OCR interaction analysis. **b** The relationship between the number of interactions per gene promoter and expression of the corresponding gene is shown. Gene expression was binned into the lowest 20th, 20–40th, 40–60th, 60–80th, and >80th percentiles. Lower and upper boxplot hinges correspond to the first and third quartiles, and outliers were defined as >1.5*IQR from the hinge. WashU browser depictions of fragment-based promoter interactions and ATAC-seq accessibility at the *IL21* (**c**) and *IFNG* (**d**) loci in TFH (red) and naive T cells (blue). Color gradients represent the CHiCAGO scores with a lower threshold of 5.

promoter-interacting OCR as iOCR. Roughly half of iOCR (15,109) are located in intergenic or intronic regions relatively far from promoters, while the other 16,295 promoter-interacting OCR are located in the promoters of other genes. The distance between promoter and promoter-interacting OCR pairs ranged from a few hundred base pairs to over a megabase, with a median of ~112 kb for both categories. While OCR in general are enriched for active chromatin marks[10,12], we find that iOCR are enriched up to 14-fold more for enhancer signatures compared to OCR not contacting a promoter (Fisher test $p < 2 \times 10^{-16}$, Fig. 4b and Supplementary Fig. 4b). Chromatin state modeling (chromoHMM[18]) revealed that iOCR are enriched at active promoters, bivalent promoters, and active enhancers, as defined by histone modification ChIP-seq in both naive and TFH cells[19–21] (Fig. 4c and Supplementary Fig. 4c). Open promoter regions involved in promoter interactions (prOCR) were more enriched for active promoter signatures, while promoter-interacting OCR located in intergenic/intronic space (nonprOCR) were more specifically enriched at poised and active enhancers (Fig. 4c). These results

indicate that promoter-connected OCR are biochemically distinct from non-connected OCR, and that this approach enriches for genomic elements actively engaged in gene regulation.

Using this 3D chromatin-promoter interaction landscape, we were able to connect the promoters of 18,669 genes (associated with 79,330 transcripts) to their potential regulatory elements, representing 145,568 distinct gene-OCR interactions. Roughly half (47%, 68,229) of these occur in both cell types, while 24% (34,928) occur uniquely in naive cells, and 29% (42,411) are only found in TFH (Fig. 5a). Overall, 91% of OCR-connected genes (17,021) exhibit at least one differential promoter-OCR interaction in naive vs. follicular helper T cells. The majority (82%) of OCR-connected genes were incorporated into regulatory structures consisting of more than one distal regulatory region. On average, each of these connected genes interact with 6 OCR (Supplementary Fig. 5), with 10% of connected promoters involved in 13 or more interactions. The degree of spatial connectivity exhibited by a promoter positively correlates with the level of gene expression (Fig. 5b, c), with

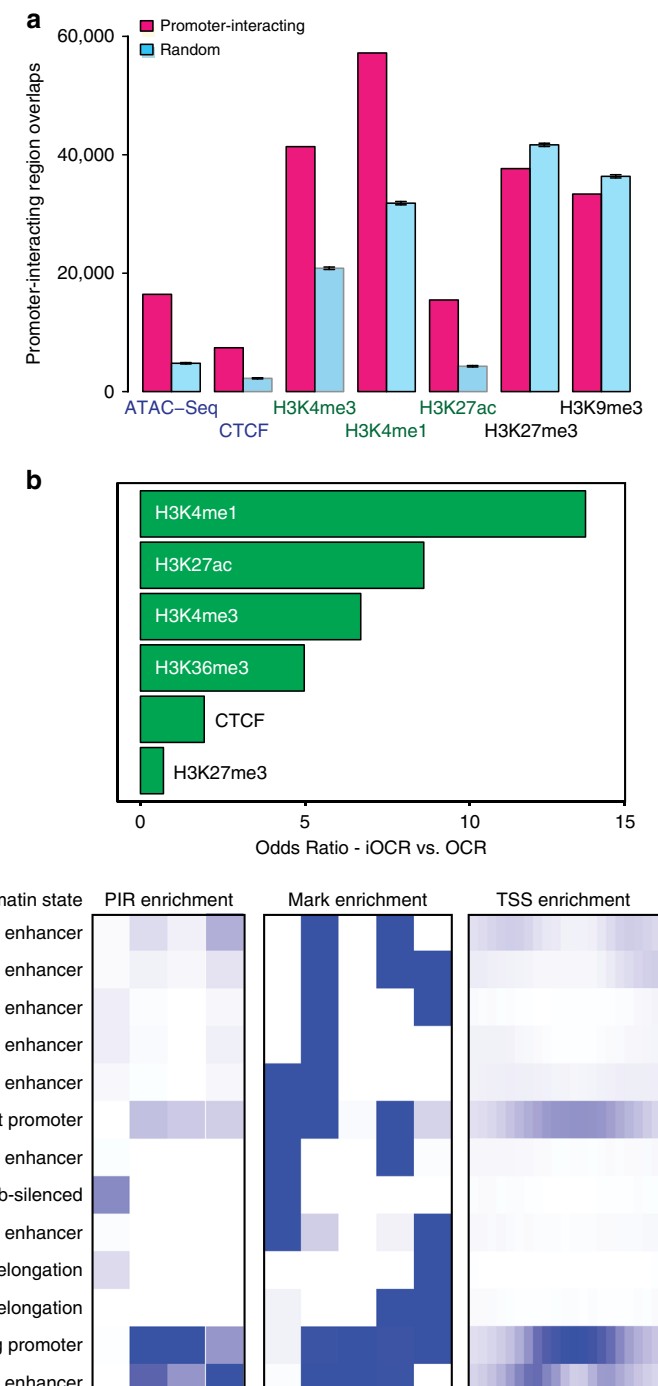

**Fig. 4 Enrichment of chromatin signatures at promoter-interacting regions. a** PIR enrichment for genomic features compared with distance-matched random regions in naive T cells using CHiCAGO. Mean ± 95% CI is depicted across 100 draws of non-significant interactions. **b** Feature enrichment at promoter-interacting OCR (iOCR) compared to a random sample of non-promoter-interacting OCR in naive T cells. **c** Enrichment of iOCR within chromHMM-defined chromatin states and TSS neighborhood in naive T cells. Fourteen ChromHMM state models based on 5 histone modifications (H3K4me1, H3K4me3, H3K27me3, H3K27ac, and H3K36me3) are shown in the middle panel, with blue color intensity representing the probability of observing the mark in each state. The heatmap to the left of the emission parameters displays the overlap fold enrichment for iOCR in promoters (prOCR) and non-promoter iOCR (nonprOCR), while the heatmap to the right shows the fold enrichment for each state within 2 kb around a set of TSS. Blue color intensity represents fold enrichment.

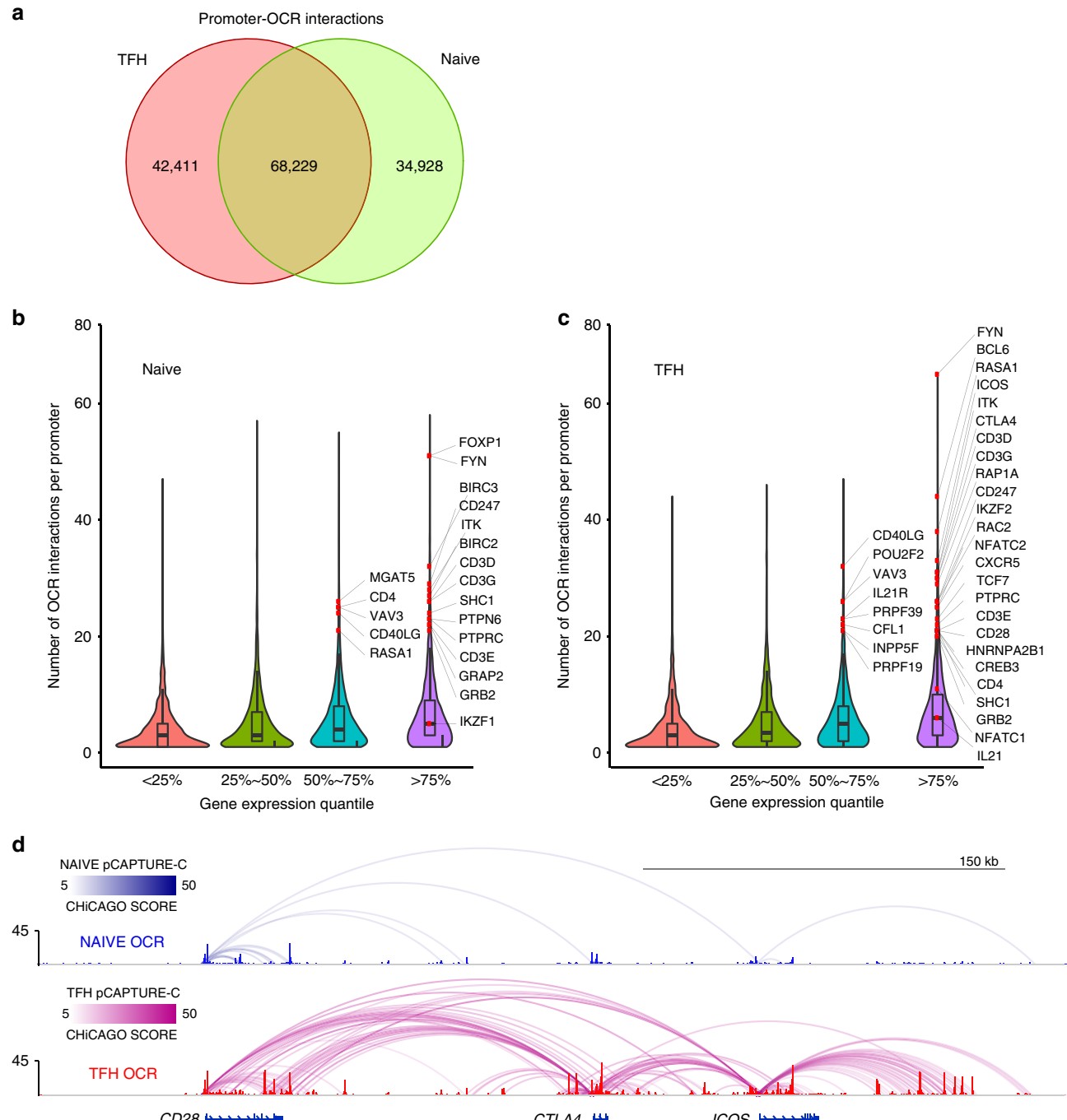

**Fig. 5 Analysis of promoter-open chromatin connectomes in naive and follicular helper T cells. a** Comparison of promoter-OCR interactions between TFH and naive T cells. Relationship between the number of promoter-OCR interactions at a gene and its corresponding expression level in naive T cells (**b**) and TFH (**c**) are depicted with genes with recognized functions in naive and TFH labeled. **d** Promoter-OCR interactions and ATAC-seq accessibility at the CD28, CTLA4, and ICOS loci in TFH (red) and naive T cells (blue). A CHiCAGO score lower threshold of 5 was used.

highly-connected promoters common to both cell types associated with genes involved in cell cycle, DNA organization and repair, protein and RNA biogenesis, and TCR signaling (Supplementary Fig. 6). Highly interactive genes in naive cells are involved in quiescence, signal transduction and immune function (Fig. 5b), while gene promoters that exhibit complex regulatory architectures in TFH are highly expressed in TFH and are involved in TFH and T cell differentiation, survival, homing, and function (Fig. 5c). For example, in naive CD4 +T cells that express CD28 but not CTLA4 or ICOS, the CD28 promoter is engaged in interactions with 8 regions of open

chromatin (Fig. 5d, blue, Supplementary Fig. 3c–e), while the CTLA4 and ICOS promoters are much less interactive. In TFH, which express all three genes, the CD28, CTLA4, and ICOS promoters adopt extensive, de novo spatial conformations involving contacts with >25 putative regulatory elements (Fig. 5d, red). Together, these results reveal major restructuring of the T cell gene regulatory architecture as naive helper cells differentiate into TFH, and indicate that complex, 3D regulatory architectures are a feature of highly expressed, lineage-specific genes involved in specialized immune functions in these disease-relevant cell types.

**Disease-associated variant-to-gene mapping for SLE.** The open chromatin landscape of TFH from the three healthy tonsils studied contained 393 accessible genomic regions that harbor SLE disease variants ($r^2 > 0.4$, Supplementary Data 3). While one-third of accessible variants (132 proxy SNPs) reside in promoters, two-thirds of accessible SLE SNPs (261) are in non-promoter regions, and how these non-promoter variants control which genes is not clear from one-dimensional epigenomic data. LDSR analysis showed very strong enrichment of SLE SNPs in promoter-connected OCR in TFH compared to naive cells (20.3 vs. 15.4, Supplementary Table 1), therefore we explored the 3D cis-regulatory architecture of TFH cells to determine whether spatial proximity of a gene to an open SLE SNP is a predictor of its role in disease.

3D variant-to-gene maps connected 256 open SLE variants (69 sentinels) to 330 potential target genes (1107 SNP-target gene pairs). Only ~9% (22) of the SLE variants that reside in TFH open chromatin interact exclusively with the nearest gene promoter (Supplementary Data 4). An example is rs35593987, a proxy to the SLE sentinel SNP rs11889341 and rs4274624 that resides in a TFH OCR and loops ~99 kb to interact with the *STAT4* promoter (Fig. 6a, Supplementary Fig. 3f). Another ~30% (75) of open SLE variants interact with nearest promoter, but also with the promoters of distant genes. An example is rs112677036, a proxy to the SLE sentinel SNP rs12938617 that resides in the first intron of *IKZF3*, interacts with nearby *IKZF3* promoter, but also interacts with promoters of two 157 kb upstream genes *PGAP3* and *ERBB2* (Fig. 6b, Supplementary Fig. 3g, h). Remarkably, over 60% of all open SLE variants (159) 'skip' the nearest gene to interact with at least one distant promoter (Supplementary Data 4). Examples of this most abundant category are rs34631447, a proxy to SLE sentinel rs6762714 residing in open chromatin in the sixth intron of the *LPP* locus, and the intergenic rs527619 and rs71041848 proxies to sentinel rs4639966 at the TREH locus. Our 3D regulatory map in TFH demonstrates that the '*LPP*' variant, in fact, does not interact with the *LPP* promoter, but instead is incorporated into a chromosomal loop structure spanning over 1 Mb that positions it in direct, spatial proximity to the promoter of *BCL6*, the 'master' transcription factor of follicular helper T cells[22–26] (Fig. 6c, Supplementary Fig. 3i). The OCR containing SLE proxies rs527619 and rs71041848 does not contact the nearby TREH gene, but instead loops 200 kb to interact with the promoter of the TFH chemokine receptor gene *CXCR5* (Fig. 6d, Supplementary Fig. 3j). Other examples of this class are rs3117582 and rs7769961, proxies to sentinel SNPs rs1150757 and rs9462027, respectively, found in contact with *LSM2* and *SNRPC* (Supplementary Fig. 7, Supplementary Fig. 3k, l). Both of these genes encode nuclear proteins that are frequently the targets of autoantibodies produced by patients with SLE[27,28].

Ontology of the set of genes found physically connected to open SLE variants were enriched for pathways involved in dendritic cell maturation, T-B cell interactions, T helper differentiation, NFkB signaling, and costimulation through CD28, ICOS, and CD40 (Fig. 7a). The top three disease networks enriched in SLE SNP-connected genes are systemic autoimmune disorders, rheumatic disease, and type 1 diabetes, all inflammatory disorders involving autoantibody-mediated pathology (Fig. 7b). At least 200 of these connected genes are differentially expressed between naive and follicular helper T cells (Supplementary Data 2 and 4), and many have known roles in TFH and/or T cell function. Similarly, SLE SNP-connected genes are highly regulated ($P < 10^{-6}$) in a hierarchical manner by IFNg, IL-2, IL-21, IL-1, IL-27, CD40L, and TCR/CD28 (Fig. 7c).

We also compared our list of genes found physically associated with SLE SNPs in TFH with those found statistically associated with SLE variants through eQTL studies in two distinct human subject cohorts. One study by Odhams et al. co-localized 97 gene-SNP eQTL in B-LCL lines[29], while another by Bentham et al. used RTC scoring in B-LCL lines and leukocytes from peripheral blood to identify 41 SLE eQTL[7]. Fully 34% (14/41) of the SNP-gene associations implicated by the Bentham study were identified by our promoter-Capture-C map in tonsillar TFH cells from three healthy donors (*ANKS1A, C6orf106, RMI2, SOCS1, PXK, UHRF1BP1, LYST, NADSYN1, DHCR7, C15orf39, MPI, CSK, ULK3, FAM219B*; Supplementary Fig. 8). Similarly, 13% (13/97) of the genes implicated by Odhams et al. were found connected to the same SLE SNPs in TFH cells (*LYST, NADSYN1, DHCR7, C15orf39, MPI, CSK, ULK3, FAM219B, TNIP1, CCDC69, SPRED2, RP11*, Supplementary Fig. 8), for a total of 16% (19/119) of SNP-gene pairs in common. Empirical distribution hypothesis testing determined that this overlap is 10-fold greater than would be expected at random ($p \sim = 0$, see "Methods"). These results indicate that a gene's spatial proximity to an accessible, disease-associated SNP in 3D is a strong predictor of its role in the context of both normal TFH biology and SLE disease pathogenesis.

**GWAS- and Capture-C-implicated OCR regulate major TFH genes.** To validate that genomic regions implicated by coalescence of ATAC-seq, promoter-focused Capture-C, and SLE GWAS signals function as bona fide distal regulatory elements for their connected promoters, we used CRISPR/CAS9 to specifically delete several iOCR harboring SLE variants from the Jurkat T cell genome. We first targeted the intergenic region near the *TREH* gene that harbors the rs527619 and rs71041848 proxies to the rs4639966 sentinel captured interacting with the *CXCR5* promoter (Supplementary Fig. 9). Neither untargeted parental Jurkat cells nor a control-targeted Jurkat line express CXCR5, but deletion of this region led to induction of CXCR5 in approximately half of the cells (Fig. 8a). Similarly, parental and control-targeted Jurkat cells do not express *IKZF1*, which encodes the transcription factor Ikaros, but deletion of the OCR containing the rs4385425 proxy to the sentinel rs11185603 (Supplementary Fig. 9) induced expression of Ikaros in nearly half of the cells (Fig. 8b). We also targeted the TFH-specific open chromatin region in *LPP* (Supplementary Fig. 9) that harbors the rs34631447 and rs79044630 proxies to sentinel rs6762714 captured interacting with the promoter of *BCL6*. *BCL6* is not expressed by parental or control-targeted Jurkat cells, but is induced by IFN-gamma (Fig. 8c). However, expression of *BCL6* was completely abrogated in Jurkat cells lacking the ~150 bp SLE-associated LPP OCR (Fig. 8c). These results confirm that SLE-associated OCR located hundreds to thousands of kilobases away in 1D interact with and act as crucial regulatory elements for the genes encoding the master TFH transcription factor *BCL6*, the *IKZF1* transcriptional repressor, and the TFH chemokine receptor *CXCR5*. These results indicate that the 3D promoter connectomes detected in these cells reveal bona fide gene regulatory architectures.

**SLE SNP connectomes tag genes involved in TFH function.** From the map of promoter-connected SLE variants in TFH cells, we noted a subset that skipped nearby promoters to interact with genes upregulated in TFH, but have no known specific role in TFH biology or SLE. These genes are enriched in canonical pathways such as mannose degradation (*MPI*), epoxysqualene biosynthesis (*FDFT1*), di- and tri-acylglycerol biosynthesis (*LCLAT1, AGPAT1*), cholesterol biosynthesis (*DHCR7, FDFT1*), oxidized GTP/dGTP detoxification (*DDX6*), breast and lung carcinoma signaling (*ERRBB2, HRAS, RASSF5, CDKN1B*), tRNA splicing (*TSEN15, PDE4A*), pentose phosphate pathway (*TALDO1*), acetyl-coa biosynthesis (*PDHB*), dolichyl-

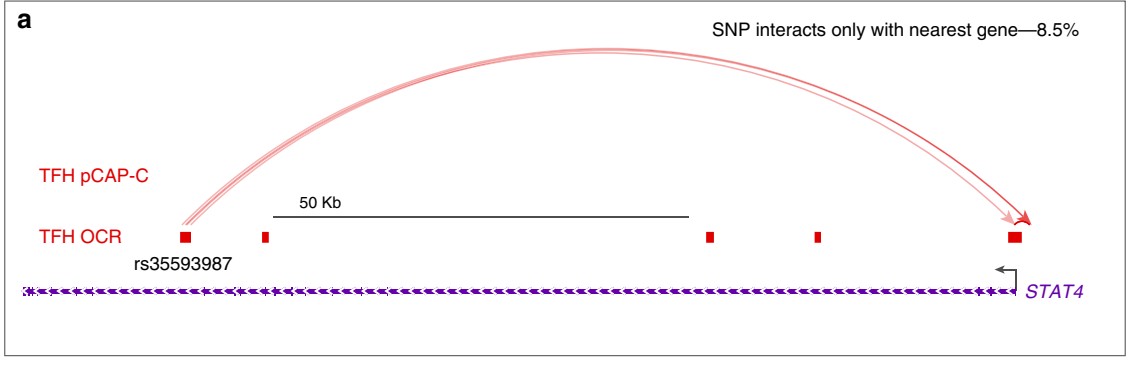

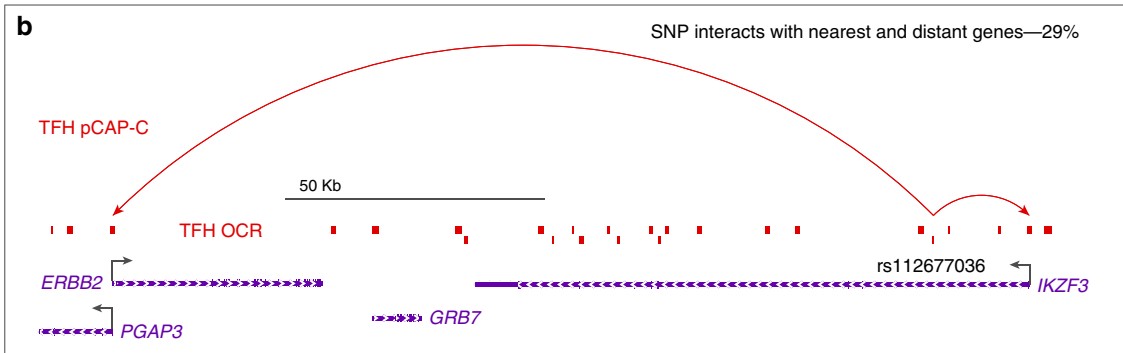

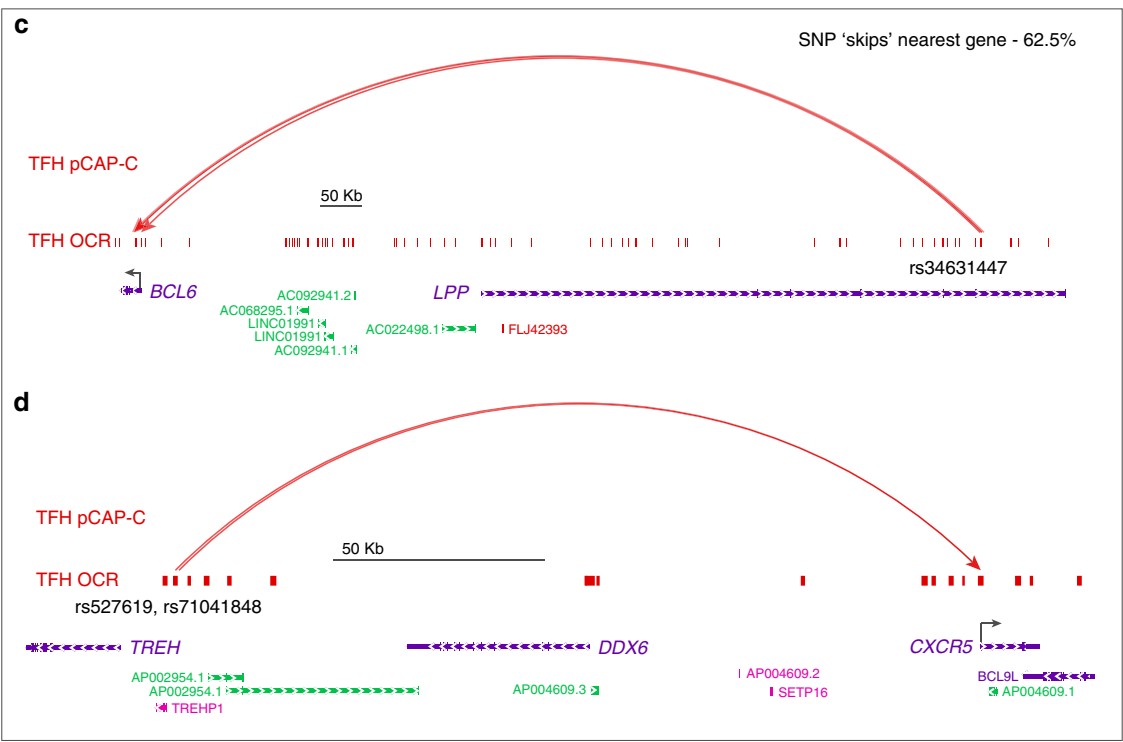

**Fig. 6 SLE variant-to-gene mapping through integration of GWAS and promoter-open chromatin connectomes in follicular helper T cells.** Four categories of accessible SLE SNP-promoter interactions were detected. **a** Accessible SLE SNP uniquely interacts with the nearest promoter (8.5%). An example is rs35593987 ($r^2 = 0.68$), which physically interacts only with the nearest gene *STAT4*. **b** Accessible SLE SNP interacts with the nearest promoter and at least one distant promoter (29%). An example is rs112677036 ($r^2 = 0.93$), which physically interacts with *IKZF3* and the distant *ERBB2* and *PGAP3* genes. **c** Accessible SLE SNP 'skips' the nearest promoter to interact exclusively with one or more distant promoters (62.5%). Examples are **c** rs34631447 ($r^2 = 1.0$), which skips *LPP* to physically interact with *BCL6*, and **d** rs527619 ($r^2 = 0.51$) and rs71041848 ($r^2 = 0.44$), which interact with the distant CXCR5 gene instead of the nearest gene, TREH. All interactions are based upon a CHiCAGO score lower threshold of 5.

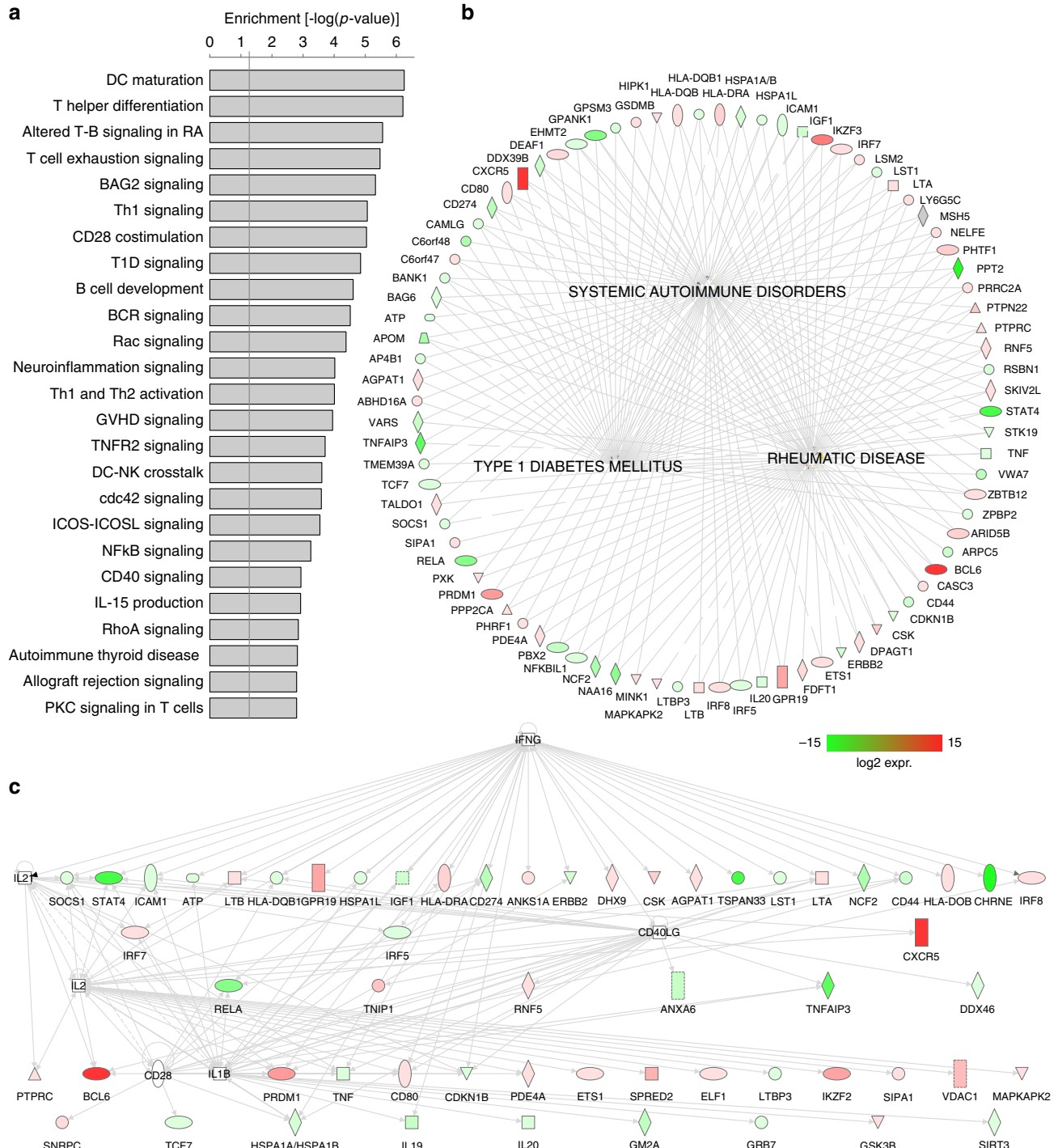

**Fig. 7 Ontology and pathway analysis of genes implicated through integration of GWAS and promoter-open chromatin connectomes in follicular helper T cells from human tonsil. a** Enrichment of the top 25 canonical pathways (**a**) and 3 disease networks (**b**) among genes implicated through promoter-open chromatin connectomes in TFH as measured by one-sided Fisher exact test in the IPA platform. **c** Regulatory hierarchical network from SLE-connectome-implicated genes. Color gradients in **b**, **c** represent log2 expression changes between TFH and naive T cells, with green indicating downregulation and red indicating upregulation in TFH. Blue nodes in **c** represent regulatory hubs for genes with no SLE-OCR connectome detected.

diphosphooligosaccharide biosynthesis (*DPAGT1*), and valine degradation (*HIBADH*). Two of these genes, *HIPK1* and *MINK1* (Fig. 9a, Supplementary Fig. 3m, n), encode a homeobox-interacting kinase and a MAP3/4K homolog that each regulate gene expression in other cell types[30,31]. Both *HIPK1* and *MINK1* are upregulated in TFH, and their promoters interact with OCR that are genetically associated with SLE risk, suggesting they are involved in TFH function. To test this, we transduced TFH

differentiated in vitro from naive CD4+ T cells[32] (Fig. 9b, c) with a lentiviral vector expressing shRNA targeting the *HIPK1* transcript (Fig. 9d), or with scrambled or *B2M* shRNAs as controls (Supplementary Fig. 10). Transduced cells were sorted, restimulated with CD3/28 beads, and secretion of IL-21, the major cytokine required for T cell help for B cell antibody production, was measured in the supernatant by ELISA. Targeting of HIPK1 expression had no effect on in vitro TFH

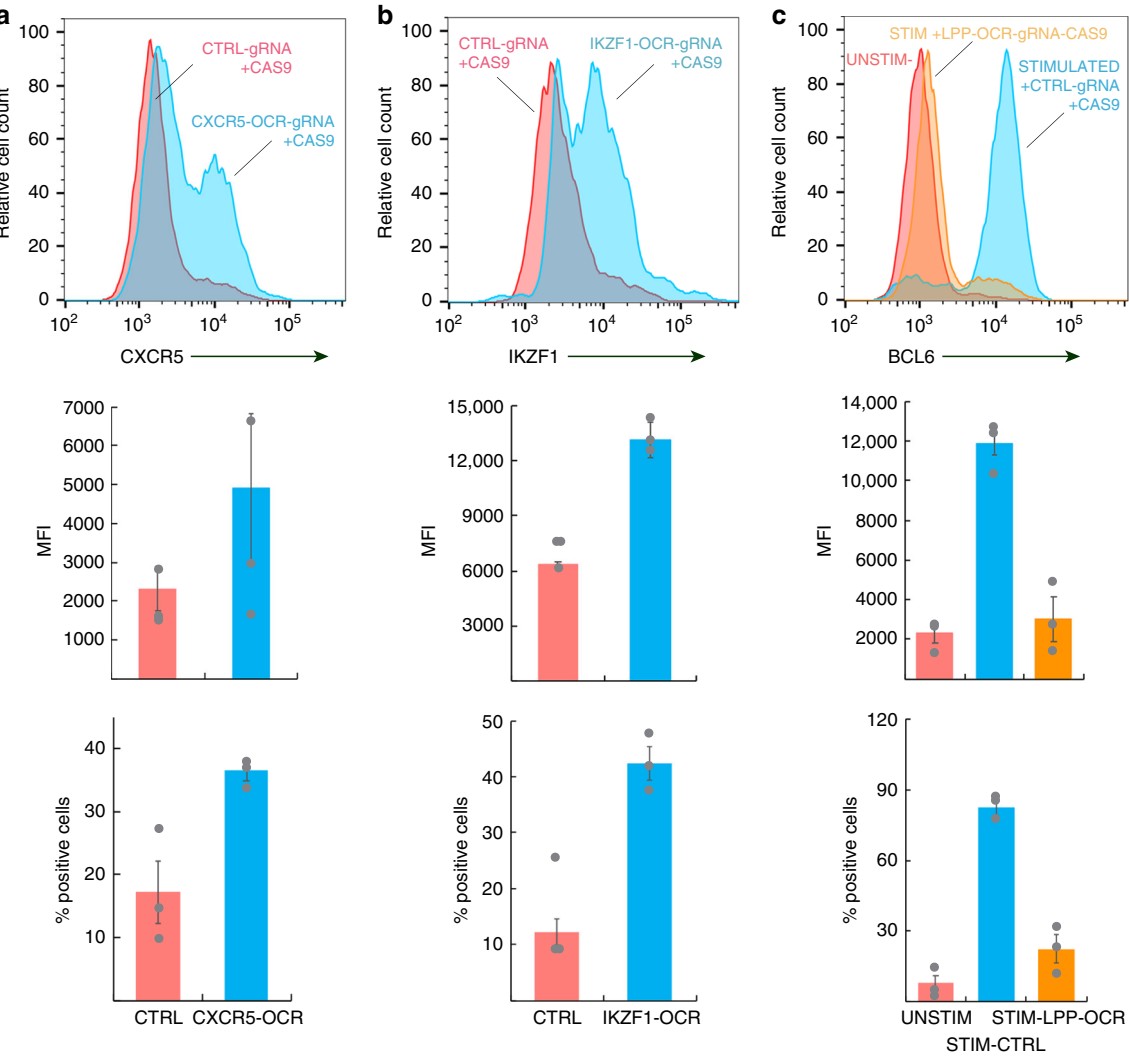

**Fig. 8 CRISPR/CAS9-based deletion of accessible, promoter-connected genomic regions harboring SLE variants influences TFH gene expression.**
**a**, CRISPR–CAS9 targeting of the 136 bp OCR near the TREH gene harboring the rs527619 ($r^2 = 0.51$) and rs71041848 ($r^2 = 0.44$) SLE proxy SNPs and captured interacting with the CXCR5 promoter leads to increased CXCR5 expression (blue histogram) by Jurkat cells compared to cells transduced with a CTRL-gRNA+CAS9 (pink histogram). **b** CRISPR–CAS9 targeting of the 213 bp OCR containing the rs4385425 SLE proxy ($r^2 = 0.99$) to rs11185603 and captured interacting with the IKZF1 promoter leads to increased IKZF1 (Ikaros) expression (blue histogram) by Jurkat cells compared to cells transduced with a CTRL-gRNA+CAS9 (pink histogram). **c** CRISPR–CAS9 targeting of the LPP SLE proxies rs34631447 ($r^2 = 1.0$) and rs79044630 ($r^2 = 0.8$) captured interacting with the BCL6 promoter abrogates IFNg-induced BCL-6 expression (orange histogram) compared to cells transduced with a CTRL-gRNA +CAS9 (blue histogram). The red histogram shows BCL-6 expression by unstimulated Jurkat cells. Graphs in **a–c** depict biological replicate (circles) and mean (bars) fluorescence intensities (upper panels) and biological replicate (circles) and mean (bars) percent positive cells (lower panels) ±SEM (error bars) for CXCR5, Ikaros, and BCL-6 in control gRNA-tranduced vs. targeted cells. All data are representative of three independent experiments. See Supplementary Fig. 7 for design and validation of CRISPR/CAS9-mediated deletion and mutation. Source data are provided as a Source Data file.

differentiation as measured by induction of BCL6, PD-1 or CXCR5 (Supplementary Fig. 10c), but resulted in a 3-fold decrease in IL-21 production (Fig. 9e). To determine if pharmacologic targeting of HIPK1 also modulates TFH function, we treated in vitro-differentiated TFH with the HIPK1/2 inhibitor A64. Pharmacologic inhibition of HIPK activity likewise resulted in dose-dependent reduction in IL-21 production by activated TFH (Fig. 9f) without effecting proliferation, viability, or differentiation (Supplementary Fig. 10c, d). As a further test of whether SLE variant-to-gene mapping can reveal drug targets for TFH function, we targeted MINK1 pharmacologically with the MAP3/4K antagonist PF06260933. This inhibitor led to a dose-dependent reduction in IL-21 secretion by TFH cells, with an ED$_{50}$ of 5 nM (Supplementary Fig. 10e). Unlike the HIPK1 inhibitor, this MINK1 inhibitor did impact T cell IL-2

production and proliferation, but with an ED$_{50}$ 8- to 10-fold higher than its effect on IL-21 production (Supplementary Fig. 10f). IL-21 is absolutely required for SLE in mouse models, and SLE patients have elevated levels of IL-21 produced by a pathogenic subset of TFH that drives plasma cell differentiation[33,34]. To further explore relevance to SLE, we assessed the role of HIPK in the expression of a panel of genes previously associated genetically and/or experimentally with SLE[4]. From this panel of 25 genes (see "Methods"), 8 genes (IL21, PCDC1, IL6R, IL2RB, BACH2, SPRED2, ARID5B, PTPN22) were significantly inhibited by treatment of TFH cells with the HIPK antagonist (Fig. 9g). These results suggest that HIPK1 operates upstream of multiple genes involved in TFH function and the pathophysiology of SLE. These data show that integrated, 3-dimensional maps of disease-associated genetic variation, open

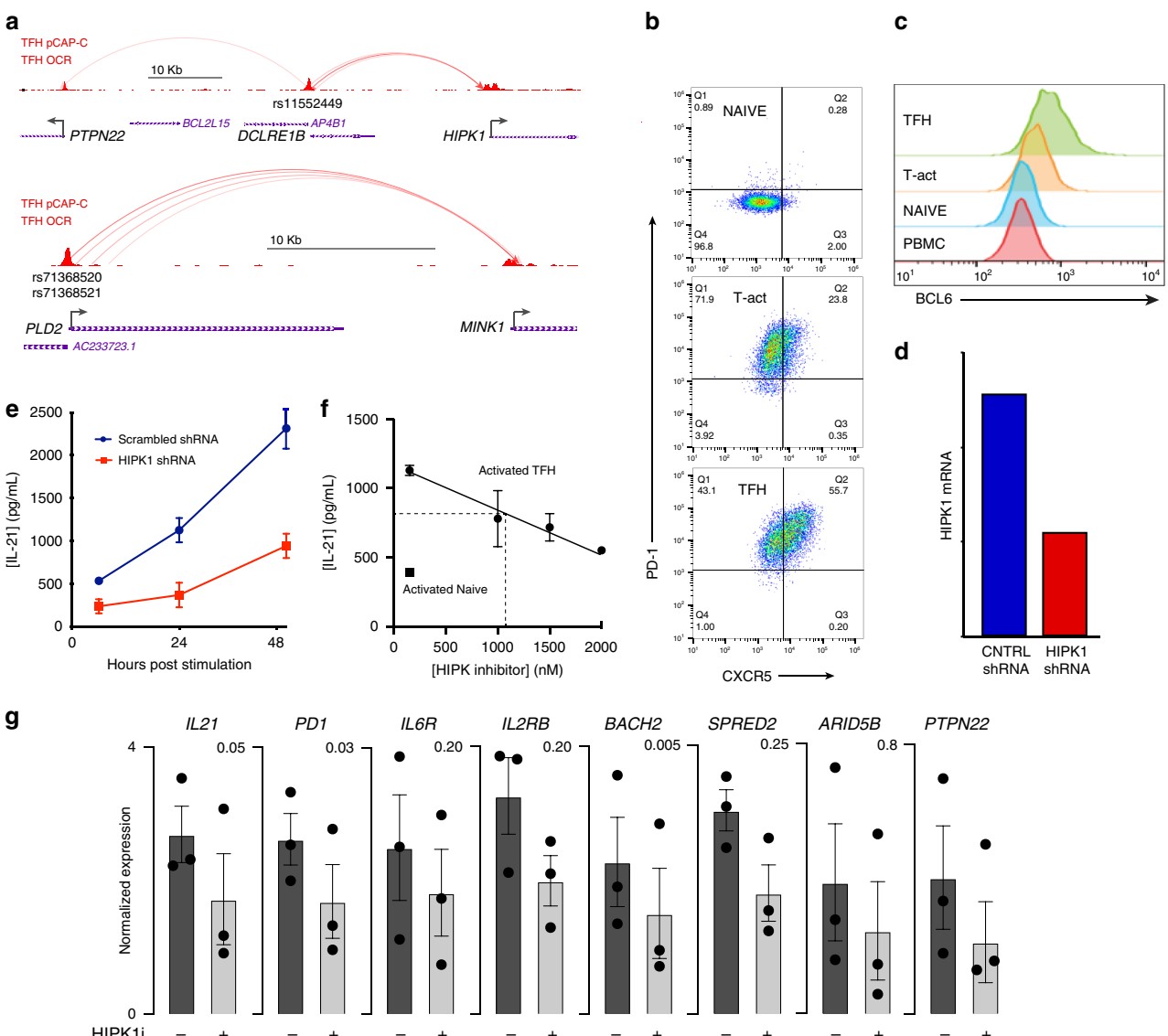

**Fig. 9 SLE variant-to-gene mapping implicates drug targets for modulation of TFH function. a** The interactomes of SLE proxies rs11552449 ($r^2 = 0.97$), rs71368520 ($r^2 = 0.5$), and rs71368521 ($r^2 = 0.5$) implicate HIPK1 and MINK1. **b** Purified naive CD4+ T cells cultured under neutral (Tact) or TFH-skewing (TFH) conditions show increased expression of PD-1 and CXCR5, as well as BCL-6 (**c**). **d** HIPK1 mRNA expression by in vitro-differentiated TFH cells transduced with scrambled Lenti-shRNA or Lenti-HIPK-1 shRNA. **e** shRNA-mediated knock-down of HIPK1 inhibits IL-21 secretion by TFH cells. A HIPK inhibitory drug causes dose-dependent inhibition of IL-21 production (**f**) and expression of several SLE-associated gene (**g**) by TFH cells. All data in e and f are mean ± SEM and are representative of three independent experiments. Data in **g** depict the mean (columns) ± SEM (error bars) normalized gene expression by three individual donor T cell cultures (circles). Statistical significance of the effect of HIPK inhibition was determined by $t$-test (two-tailed). $P$ values are: BACH2, 0.01; SPRED2, 0.02; IL2RB, 0.03; ARID5B, 0.03; PD1, 0.04; IL6R, 0.04; PTPN22, 0.05; IL21, 0.06. Source data are provided as a Source data file.

chromatin, and promoter connectomes can lead to *bona fide* drug targets that control tissue-specific and SLE-relevant biology.

## Discussion

We generated comprehensive, high-resolution maps of the open chromatin-promoter connectome of human TFH cells, allowing physical assignment of non-coding OCR and SNPs to genes, and revealing the potential regulatory architectures of nearly all coding genes and over half of non-coding genes in an immune cell type with crucial roles in humoral immunity and systemic immunopathology. This study did not determine the impact of SLE-associated genetic variation on chromatin accessibility or promoter interactions. Rather, we use the location of reported SLE variants as 'signposts' to identify open chromatin elements that may regulate SLE-relevant gene expression in the context of normal follicular helper T cell biology. Similar to previous promoter interactome studies[35], we found that promoter-interacting regions are enriched for the chromatin-based signatures of enhancers. We also find that promoter-connected OCR are enriched over 10-fold for enhancer marks compared to non-connected OCR, suggesting that promoter-focused Capture-C identifies non-coding regions with gene regulatory potential. We also observed enrichment of enhancer marks at promoter-promoter inter-actions, suggesting that some promoters may synergize in an enhancer-like manner to augment expression of their con-nected genes[36].

Similar to previous estimates[37], we find that less than 10% of promoter interactions exclusively involve the nearest genes. Over 90% of accessible disease variants interact with distant genes, and over 60% of open variants skip the nearest gene altogether and interact only with distant genes. Importantly, we were able to validate direct roles for several SLE-associated distal OCR in the regulation of their connected genes (BCL6, CXCR5, IKZF1) using CRISPR/CAS9-mediated editing in human T cells, confirming that SLE-associated genetic variation marks regulatory elements for genes with known roles in TFH and/or SLE biology. A locus control region ~130 kb upstream of BCL6 has been defined previously in germinal center B cells[38], and we find evidence for usage of this region by human TFH cells at the level of open chromatin, histone marks, and long-range connectivity to the BCL6 promoter (Supplementary Fig. 11). However, we also observe a much more distant 'stretch' enhancer within the LPP gene in TFH cells, as evidenced by extensive open chromatin, H3K27 acetylation, and H3K4 mono-methylation (Supplementary Fig. 11). This region shows extensive connectivity with BCL6 in the 3D architecture of the nucleus, and the 1 Mb distal SLE-associated BCL6 enhancer functionally validated in this study is contained within this BCL6 stretch enhancer. This enhancer region is occupied in lymphoid cell lines by NFkB/RelA and POU2F2, both of which positively regulate immunoglobulin and inflammatory gene expression. Long-range regulatory elements for CXCR5 have not been previously identified, and the -180 kb SLE-associated element is the first validated for CXCR5. Deletion of this element led to increased expression of CXCR5, suggesting that unlike the distal BCL6-LPP enhancer, this element is a silencer in Jurkat cells. Consistent with this finding, this region is occupied by the repressive transcription factors YY1, BHLHE40, and BATF in lymphoid cells (ENCODE), but its function in primary TFH cells remains to be determined. These distant SNP-gene regulatory pairs join examples like the FTO and TCF7L2 loci[1,3], in which GWAS data were interpreted to implicate the nearest genes, while 3D epigenomics and functional follow-up showed that the disease variants reside in elements that regulate distant genes. Our results indicate that a gene's spatial proximity in three dimensions to a regulatory SNP is a strong predictor of its function in the context of TFH biology and SLE disease pathogenesis, and suggest assumptions that a given genomic or epigenomic feature identified by 1D mapping regulates the nearest gene are more likely to be incorrect than correct.

Our physical variant-to-gene maps in tonsillar TFH cells from three healthy individuals identified a large proportion of the SNP-gene pairs detected statistically in SLE eQTL studies[7,29]. These quantitative trait studies require samples from hundreds of individuals, and the data are obtained from blood, B-LCL, or naive mononuclear leukocytes. However, immune responses do not take place in the blood, and the pathophysiologic aspects of inflammatory disease are mediated by specialized, differentiated immune cell types that are rare or not present in blood. Our approach utilized human follicular helper T cells from tonsil that are 'caught in the act' of mediating in vivo T-B immune responses, and is the same cell type involved in B cell help for autoantibody production in SLE. This variant-to-gene mapping approach identified ~10-fold more SLE SNP-gene associations than current eQTL studies; further follow up work will determine how many of these associations are valid vs. false positives.

In addition to revealing the previously unknown SLE-associated regulatory architectures of known TFH/SLE genes, we show that the combination of GWAS and 3D epigenomics can identify genes with previously unappreciated roles in disease biology. In a previous study, we implicated the gene ING3 by virtue of its interaction with an osteoporosis-associated SNP, and showed that this gene is required for osteoclast differentiation in an in vitro

model[15]. In this current study, approximately two dozen 'novel' genes up- or downregulated during TFH differentiation were implicated through their connection to SLE SNPs. Among these are HIPK1, a nuclear homeobox-interacting protein kinase that cooperates with homeobox, p53, and TGFB/Wnt pathway transcription factors to regulate gene transcription[30,39–41]. A role for HIPK1 in T-independent B cell responses has be identified in the mouse[42], but no role for this kinase has been previously established in TFH or SLE. Another gene implicated in our study is MINK1, which encodes the misshapen-like kinase MAP4K6. This kinase functions upstream of JNK and SMAD in neurons[43,44], and has been shown to inhibit TGFB-induced Th17 differentiation[45]. However, a role in TFH or SLE has likewise not been previously appreciated. We show that genetic and/or pharmacologic targeting of HIPK1 or MINK1 in human TFH cells inhibits their production of IL-21, a cytokine required for T cell-mediated help for B cell antibody production with specific links to autoantibody production in human SLE patients[46]. While further work is required to elucidate the role of these kinases in TFH biology and the pathogenesis of systemic autoimmunity, these examples show the utility of this integrated approach in identifying targets for drug repurposing or compound development in complex heritable diseases.

## Methods

**Antibodies**. Anti-human APC-BCL-6 (Biolegend, cat # 358506, 5 μl/test) after treatment with human recombinant IFNg (5 ng/mL, R&D Systems, cat # 285-IF) overnight and stimulation with PMA (30 ng/mL, cat # P-8139) and ionomycin (1 μM, Sigma-Aldrich, cat # I-0634) for 4–6 h. Expression of Ikaros and CXCR5 by targeted Jurkat lines was also assessed by flow cytometry using anti-human APC-CXCR5 (Biolegend, cat # 356908, 5 μl/test) and anti-human PE-Ikaros (BD Biosciences, cat # 564476, 5 μl/test). Biolegend antibodies Anti-human PE/Dazzle 594-CXCR5 (5 μl/test) cat # 356928 and Anti-human Brilliant Violet 421-PD-1 (5 μl/test) cat # 329920 were used for the human T cells staining. Biolegend antibodies against CD4 APC-Cy7 (1/100 dilution, cat. 317418), CD45RO Pac Blue (1/50 dilution, cat. 304216), CD25 PE (1/25 dilution, cat. 302606), CXCR5 APC (1/10 dilution, cat. 356908), PD1 PeCy7 (1/50 dilution, cat. 329918) were used for tonsillar T cell sorting.

**Purification of helper T cells from human tonsil**. Fresh tonsils were obtained as discarded surgical waste from de-identified immune-competent children (n = 10) undergoing tonsillectomy to address airway obstruction or a history of recurrent tonsillitis. These studies were approved by The Children's Hospital of Philadelphia Institutional Review Board as non-human subjects research. The mean age of donors was 5.7 years (range 2–16 years) and 50% were male. Tonsillar mononuclear cells were isolated from tissues by mechanical disruption (tonsils were minced and pressed through a 70-micron cell screen) followed by Ficoll-Paque centrifugation. CD19-positive cells were removed (StemCell) and CD4+ T cells were enriched with magnetic beads (Biolegend) prior to sorting naive T cells (CD4 +CD45RO-) and T follicular helper cells (CD4+CD45RO+CD25loCXCR5hiPD1hi) on a MoFlo Astrios EQ (Beckman Coulter). The gating strategy is shown in Supplementary Fig. 12.

**Cell fixation**. We used standard methods for cell fixation[9]. Briefly, 10^7 TFH or naive CD4+ T cells were suspended in 10 mL RPMI + 10% FBS, followed by an additional 270 μL of 37% formaldehyde and incubation for 10 min at RT on a platform rocker. The fixation reaction was quenched by the addition of 1.5 mL cold 1 M glycine (4 °C). Fixed cells were centrifuged at 210 × g for 5 min at 4 °C and supernatants were removed. The cell pellets were washed in 10 ml cold PBS (4 °C) followed by centrifugation as above. Cell pellets were resuspended in 5 ml cold lysis buffer (10 mM Tris pH8, 10 mM NaCl, 0.2% NP-40/Igepal supplemented with a protease inhibitor cocktail). Resuspended cell pellets were incubated for 20 min on ice, centrifuged at 680 × g, and lysis buffer was removed. Cell pellets were resuspended in 1 mL of fresh lysis buffer, transferred to 1.5 mL Eppendorf tubes, and snap frozen in ethanol/dry ice or liquid nitrogen. Frozen cell pellets were stored at −80 °C for 3C library generation.

**3C library generation**. We used standard methods for generation of 3C libraries[15]. For each library, 10^7 fixed cells were thawed at 37 °C, followed by centrifugation at RT for 5 min at 1845 × g. The cell pellet was resuspended in 1 mL of dH2O supplemented with 5 μL 200X protease inhibitor cocktail, incubated on ice for 10 mins, then centrifuged. Cell pellet was resuspended to a total volume of 650 μL in dH2O. 50 μL of cell suspension was set aside for pre-digestion QC, and the remaining

sample was divided into 3 tubes. Both pre-digestion controls and samples underwent a pre-digestion incubation in a Thermomixer (BenchMark) with the addition of 0.3%SDS, 1× NEB DpnII restriction buffer, and dH2O for 1 h at 37 °C shaking at 1000 rpm. A 1.7% solution of Triton X-100 was added to each tube and shaking was continued for another hour. After pre-digestion incubation, 10 µl of DpnII (NEB, 50 U/µL) was added to each sample tube only, and continued shaking along with pre-digestion control until the end of the day. An additional 10 µL of DpnII was added to each digestion reaction and digested overnight. The next day, a further 10 µL DpnII was added and continue shaking for another 2–3 h. 100 µL of each digestion reaction was then removed, pooled into one 1.5 mL tube, and set aside for digestion efficiency QC. The remaining samples were heat inactivated incubated at 1000 rpm in a MultiTherm for 20 min at 65 °C to inactivate the DpnII, and cooled on ice for 20 additional minutes. Digested samples were ligated with 8 µL of T4 DNA ligase (HC ThermoFisher, 30 U/µL) and 1X ligase buffer at 1000 rpm overnight at 16 °C in a MultiTherm. The next day, an additional 2 µL of T4 DNA ligase was spiked in to each sample and incubated for another few hours. The ligated samples were then de-crosslinked overnight at 65 °C with Proteinase K (20 mg/mL, Denville Scientific) along with pre-digestion and digestion control. The following morning, both controls and ligated samples were incubated for 30 min at 37 °C with RNase A (Millipore), followed by phenol/chloroform extraction, ethanol precipitation at −20 °C, the 3C libraries were centrifuged at $85 \times g$ for 45 min at 4 °C to pellet the samples. The controls were centrifuged at $1845 \times g$. The pellets were resuspended in 70% ethanol and centrifuged as described above. The pellets of 3C libraries and controls were resuspended in 300 µL and 20 µL dH2O, respectively, and stored at −20 °C. Sample concentrations were measured by Qubit. Digestion and ligation efficiencies were assessed by gel electrophoresis on a 0.9% agarose gel and also by quantitative PCR (SYBR green, Thermo Fisher).

**Promoter-Capture-C design.** Our promoter-Capture-C approach was designed to leverage the four-cutter restriction enzyme *DpnII* in order to give high resolution restriction fragments of a median of ~250 bp[15]. This approach also allows for scalable resolution through in silico fragment concatenation (Supplementary Table 4). Custom capture baits were designed using Agilent SureSelect RNA probes targeting both ends of the *DpnII* restriction fragments containing promoters for coding mRNA, non-coding RNA, antisense RNA, snRNA, miRNA, snoRNA, and lincRNA transcripts (UCSC lincRNA transcripts and sno/miRNA under GRCh37/hg19 assembly) totaling 36,691 RNA baited fragments through the genome[9]. In this study, the capture library was re-annotated under gencodeV19 at both 1-fragment and 4-fragment resolution, and is successful in capturing 89% of all coding genes and 57% of noncoding RNA gene types. The missing coding genes could not be targeted due to duplication or highly repetitive DNA sequences in their promoter regions.

**Promoter-Capture-C assay.** Isolated DNA from 3 C libraries was quantified using a Qubit fluorometer (Life Technologies), and 10 µg of each library was sheared in dH2O using a QSonica Q800R to an average fragment size of 350 bp. QSonica settings used were 60% amplitude, 30 s on, 30 s off, 2 min intervals, for a total of 5 intervals at 4 °C. After shearing, DNA was purified using AMPureXP beads (Agencourt). DNA size was assessed on a Bioanalyzer 2100 using a DNA 1000 Chip (Agilent) and DNA concentration was checked via Qubit. SureSelect XT library prep kits (Agilent) were used to repair DNA ends and for adaptor ligation following the manufacturer protocol. Excess adaptors were removed using AMPureXP beads. Size and concentration were checked by Bioanalyzer using a DNA 1000 Chip and by Qubit fluorometer before hybridization. One microgram of adaptor-ligated library was used as input for the SureSelect XT capture kit using manufacturer protocol and our custom-designed 41K promoter Capture-C library. The quantity and quality of the captured library was assessed by Bioanalyzer0a high sensitivity DNA Chip and by Qubit fluorometer. SureSelect XT libraries were then paired-end sequenced on 8 lanes of Illumina Hiseq 4000 platform (100 bp read length).

**ATAC-seq library generation.** A total of 50,000 to 100,000 sorted tonsillar naive or follicular helper T cells were centrifuged at $550 \times g$ for 5 min at 4 °C. The cell pellet was washed with cold PBS and resuspended in 50 µL cold lysis buffer (10 mM Tris-HCl, pH 7.4, 10 mM NaCl, 3 mM $MgCl_2$, 0.1% NP-40/IGEPAL CA-630) and immediately centrifuged at $550 \times g$ for 10 min at 4 °C. Nuclei were resuspended in the Nextera transposition reaction mix (25 µl 2× TD Buffer, 2.5 µL Nextera Tn5 transposase (Illumina Cat #FC-121-1030), and 22.5 µl nuclease free $H_2O$) on ice, then incubated for 45 min at 37 °C. The tagmented DNA was then purified using the Qiagen MinElute kit eluted with 10.5 µL Elution Buffer (EB). Ten microliters of purified tagmented DNA was PCR amplified using Nextera primers for 12 cycles to generate each library. PCR reaction was subsequently cleaned up using 1.5x AMPureXP beads (Agencourt), and concentrations were measured by Qubit. Libraries were paired-end sequenced on the Illumina HiSeq 4000 platform (100 bp read length).

**ATAC-seq analysis.** TFH and naive ATAC-seq peaks were called using the ENCODE ATAC-seq pipeline (https://www.encodeproject.org/atac-seq/). Briefly, pair-end reads from three biological replicates for each cell type were aligned to

hg19 genome using bowtie2, and duplicate reads were removed from the alignment. Narrow peaks were called independently for each replicate using macs2 (-p 0.01 --nomodel --shift -75 --extsize 150 -B --SPMR --keep-dup all --call-summits) and ENCODE blacklist regions (ENCSR636HFF) were removed from peaks in individual replicates. Peaks from all replicates were merged by bedtools (v2.25.0) within each cell type and the merged peaks present in less than two biological replicates were removed from further analysis. Finally, ATAC-seq peaks from both cell types were merged to obtain reference open chromatin regions. To determine whether an OCR is present in TFH and/or naive cells, we first intersected peaks identified from individual replicates in each cell type with reference OCRs. If any peaks from at least one replicate overlapped with a given reference OCR, we consider that region is open in the originating cell type. Quantitative comparisons of TFH and naive open chromatin landscapes were performed by evaluating read count differences against the reference OCR set. De-duplicated read counts for OCR were calculated for each library and normalized against background (10 K bins of genome) using the R package csaw (v 1.8.1). OCR peaks with less than 1.5 CPM (4.5 ~ 7.5 reads) support in the top 50% of samples were removed from further differential analysis. Differential analysis was performed independently using edgeR (v 3.16.5) and limmaVoom (v 3.30.13). Differential OCR between cell types were called if FDR < 0.05 and absolute log2 fold change >1 in both methods.

**Promoter-focused Capture-C analysis.** Paired-end reads from three biological replicates for naive and follicular helper T cells were pre-processed using the HICUP pipeline[16] (v0.5.9), with bowtie2 as aligner and hg19 as the reference genome (Supplementary Data 5). We were able to detect non-hybrid reads from all targeted promoters, validating the success of the promoter capture procedure. Significant promoter interactions at 1-DpnII fragment resolution were called using CHiCAGO[17] (v1.1.8) with default parameters except for binsize set to 2500. Significant interactions at 4-DpnII fragment resolution were also called using CHiCAGO with artificial.baitmap and .rmap files in which DpnII fragments were concatenated in silico into 4 consecutive fragments using default parameters except for removeAdjacent set to False. The significant interactions (CHiCAGO score >5) from both 1-fragment and 4-fragment resolutions were exported in .ibed format and merged into a single file using custom a PERL script to remove redundant interactions and to keep the max CHiCAGO score for each interaction. Open chromatin interaction landscapes were established by projecting significant DpnII fragment interactions at merged 1- and 4-fragment resolutions to reference OCR (Fig. 3a). First, DpnII fragments involved in significant interactions (both "bait" and "other end") were intersected with reference OCR using bedtools (v2.25.0). Interactions between bait and other end OCR pairs were called independently for each cell type if their overlapped fragments interacted at either resolution and if both OCR were called as "open" in the corresponding cell type. OCR involved in promoter interactions (iOCR) were classified as promoter OCR (prOCR) or regulatory OCR (nonprOCR) by comparing their genomic locations to pre-defined promoter regions (−1500 bp ~500 bp of TSS) of transcripts in GENCODE V19 and UCSC noncoding RNA described above. If both ends of an OCR interaction failed to overlap a gene promoter, that interaction was removed. OCR pair interactions were combined from both cell types to obtain the reference open chromatin promoter-captured interaction landscapes.

**SLE GWAS data integration.** GWAS reports sentinel SNPs, i.e., those present on genotyping arrays that were associated with disease at genome-wide significance ($p < 5 \times 10^{-8}$). Proxy SNPs are those SNPs frequently co-inherited with the sentinels, but were either not present on the genotyping arrays, or were not associated with disease at genome-wide significance. The frequency of co-inheritance, or LD ($r^2$), required to categorize a proxy SNP as linked to sentinel SNP can be difficult to determine empirically. We chose a lower LD threshold ($r^2 > 0.4$) for the reasons outlined below. If we restrict our analyses to proxies in tight LD ($r^2 > 0.8$) with SLE sentinel SNPs, we detect a total of 190 proxies located in 109 OCR in tonsillar T cells, and 112 of these open proxies are connected to 193 target genes. Relaxing the LD threshold to $r^2 > 0.4$ results in detection of 432 accessible proxies in 246 OCR, and 256 open SLE variants connected to 330 target genes. Therefore, including proxies in lower LD with SLE sentinels only ~doubles the number of OCRs under investigation, and only increases the number of implicated target genes by 1.7-fold. Genes with accessible $r^2 > 0.8$ proxies in their promoters are expressed more highly than a random sample of genes with open promoters ($P \sim 4 \times 10^{-6}$). This same analysis using an $r^2 > 0.4$ threshold still shows strong enrichment for highly expressed genes at even higher statistical significance ($p \sim 5 \times 10^{-12}$). Pathway analysis of genes found connected to SLE proxies gives very similar results whether performed with >0.4 or >0.8 $r^2$ thresholds, and the top disease networks enriched in both cases are systemic autoimmune disorders, rheumatic disease, and type 1 diabetes. If we use an $r^2 > 0.8$ LD threshold to test the statistical significance of the overlap between our physical SNP-gene associations and the statistical SNP-gene associations from eQTL studies (empirical distribution hypothesis testing—see below), the expected number of overlapping sentinel-gene pairs between our Capture-C data and the eQTL studies is 1. The observed overlap in this case is 7, indicating an enrichment of 7-fold over random, with $p \sim 10^{-5}$. Using a relaxed $r^2 > 0.4$ LD threshold, the expected number of overlapping sentinel-gene pairs is 1.8, and the observed is 19, indicating a 10-fold enrichment with a $p$ value approaching zero. In each of the three examples above, if lowering

the LD threshold were merely bringing more noise (i.e., irrelevant SNPs) into the system, we would expect to see the effect sizes decrease. Because lowering the threshold increases N, statistical significance may or may not decrease. However, we find that lowering the LD threshold increases, not decreases, the effect size and significance of the associations tested. These results indicate that including SNPs in lower LD with SLE sentinels in this case actually increases the signal-to-noise ratio of the analyses reported, presumably because it includes plenty of relevant SNPs.

**Partitioned heritability LD score regression enrichment analysis.** Partitioned heritability LD Score Regression[47] (v1.0.0) was used to identify enrichment of GWAS summary statistics among open accessible regions identified from both naive T and TFH ATAC-seq. The baseline analysis was performed using LDSCORE data (https://data.broadinstitute.org/alkesgroup/LDSCORE) with LD scores, regression weights, and allele frequencies from 1000G Phase1. The summary statistics from Langefeld et al[48]. were downloaded from the GWAS catalog ftp://ftp.ebi.ac.uk/pub/databases/gwas/summary_statistics/Lange-feldCD_28714469_GCST007400, and SNPs were filtered with minor allele frequency >0.1 and imputation score >0.4. We generated partitioned LD score regression annotations for naive and TFH using the coordinates of the all OCR or iOCR (promoter OCR + promoter-interacting OCR) as previously performed[48]. Finally, the cell-type-specific partitioned LD scores were compared to baseline LD scores to measure enrichment in TFH and naive cells independently.

**Microarray and gene set enrichment analysis.** RNA from two biological naive tonsillar CD4+ T cell replicates and four biological tonsillar TFH replicates were hybridized to Affymetrix Human Clarion S arrays at the CHOP Nucleic Acid and Protein Core. Data were pre-processed (RMA normalization), and analyzed for differential expression (DE) using Transcriptome Analysis Console v 4.0 with 2-way ANOVA implementation and Benjamini-Hochber multiple testing adjustments. We used a false discovery rate (FDR) threshold of 0.05 and a fold-change (FC) threshold of 2. Lists of differentially expressed genes were generated and ranked by log2 fold change. The log2 fold change of the genes with significantly differential accessibility at promoter regions were compared to the pre-ranked gene expression data for GSEA enrichment analysis. GSEA enrichment analysis was performed to evaluate whether the set of differentially expressed genes (tfh_upreg.grp or tfh_down.grp) between TFH and naive by our microarray data were overrepresented at the top or bottom of a gene list pre-ranked by their promoter accessibility (log fold change) detected in our ATAC-seq data.

**ChromHMM, and Ingenuity pathway analysis.** Histone mark and CTCF ChIP-seq datasets for naive and follicular helper T cells were obtained from public resources[19–21] and compared to promoter-interacting fragments or promoter-interacting OCR. Enrichment of promoter-interacted fragments (PIR) for histone marks and CTCF regions was determined independently in each cell type using the function peakEnrichment4Features in the CHiCAGO package, and feature enrichment at promoter-interacting OCR was compared to enrichment at non-promoter-interacting OCR using the feature enrichment R package LOLA (v1.4.0)[49]. One-sided fisher's exact tests were performed and odd ratios were plotted for significant enrichment ($p$-value < $10^{-6}$) using ggplot2. The chromatin states of promoter-interacting OCR were also determined using ChromHMM (v1.17) on binarized bed file of histone marks ChIP-seq peaks with 15 states for naive T cells (active enhancer, active genic enhancer, poised genic enhancer, poised intergenic enhancer, bivalent intergenic enhancer, active bivalent promoter, bivalent intergenic enhancer, polycomb-silenced, weak bivalent enhancer, transcriptional elongation 1, transcriptional elongation 2, actively transcribing promoter, poised promoter or enhancer, neutral/quiescent chromatin, and 'unknown state') and 8 states (poised intergenic enhancer, bivalent intergenic enhancer, active bivalent promoter, bivalent intergenic enhancer, polycomb-silenced, weak bivalent enhancer, actively transcribing promoter, poised promoter or enhancer, and neutral/quiescent chromatin) for TFH cells. The annotation of chromatin states was manually added with the reference to epigenome roadmap project[20]. Ingenuity pathway analysis (IPA, QIAGEN) was used for all the pathway analysis. The top significantly enriched canonical pathways were plotted using ggplot2 and networks with relevant genes were directly exported from IPA.

**Empirical distribution hypothesis testing.** We addressed the statistical significance of the overlap between our physical SNP-gene associations measured in TFH cells and the statistical SNP-gene associations from prior eQTL studies. Two challenges to these comparisons should be noted: (1) Both eQTL studies used different sentinel sets from the updated set we use in this manuscript, and 2) all these studies use different cell types. The study by Odhams used a colocalization method on lymphoblastoid cell lines (LCLs), while the Bentham study used an RTC scoring method to identify eQTL among multiple cell types including NK cells, monocytes, B cells, and CD4+ T cells. Our variant-gene mapping is based on physical interactions between SNPs and genes in TFH cells. Theoretically, the same variants may regulate different genes in different cell types. However, to achieve the fairest possible statistical comparisons between these groups, we used an empirical distribution hypothesis testing approach. We identified genes within 5Mbp from

each sentinel from the most current set and used these to generate a pool of randomly-associated gene-variant pairs. We then randomly picked the same number of pairs from this random pool as we observed using our capture-C method, and combined these random pairs with the eQTL data to calculate the frequency of random overlapping gene-sentinel pairs. These steps were repeated 100,000 times to calculate an empirical distribution. We then compared the observed number of SNP-gene pair overlaps to the empirical distribution to generate a $p$-value.

**CRISPR/CAS9 genome editing.** CRISPR guide RNAs (sgRNA) targeting rs34631447, rs79044630, rs527619, rs71041848, and rs4385425 were designed using http://crispr.tefor.net and cloned into lentiCRISPRv2-puro or lentiCRISPRv2-mCherry (Addgene plasmid #52961; http://n2t.net/addgene:52961; RRID: Addgene_52961) by golden gate ligation using the *BsmB1* restriction enzyme (NEB). 293T cells were transfected in DMEM using Lipofectamine 2000 (Invitrogen) with 6 µg PsPAX2 and 3.5 µg PmD2.G packaging plasmids and 10 µg empty lentiCRISPRv2 or 10 µg sgRNA-encoding lentiCRISPRv2. Viral supernatants were collected after 48 h for transduction into Jurkat leukemic T cells maintained in RPMI 1640 with 10% fetal bovine serum, L-glutamine, 2-mercaptoethanol, and penicillin/streptomycin. Cells were seeded in a 24 well plate at $0.5 \times 10^6$ in 0.5 mL of media per well, and 1 mL of viral supernatant with 8 µg/mL of polybrene was added to each well. Spin-fection was performed for 90 min. at 2500 rpm and 25 °C, and transduced cells were equilibrated at 37C for 6 h. For rs34631447, rs79044630, and rs4385425, 1.2 ml of media was removed and replaced with 1 ml of fresh media containing 1 µg of puromycin for 7 days of selection before use in experiments. Cells transduced with sgRNAs targeting rs527619 and rs71041848 were sorted based on mCherry on a FACS Jazz (BD Biosciences). Mutations were analyzed by PCR coupled with Sanger sequencing at the CHOP Nucleic Acids and Protein Core. The following primers were used for PCR: BLC6-F: CTCTGTGGTTGTGGGCAAGGC-R:CAGGTGGCGAATCAGGA CAGG, CXCR5-F: GTCCCTGGTGATGGAAACTCAGGC-R: GCAGTGGCCTC CCTTACACAGG, IKZF1-F: CCTTCTCCATGCCCAGGTGACTC-R: GGCCT CAGCTAGGCAAACCAGAG. Measurement of BCL-6 expression in targeted Jurkat lines was assessed by flow cytometry using anti-human APC-BCL-6 (Biolegend) after treatment with human recombinant IFNg (5 ng/mL, R&D Systems) overnight and stimulation with PMA (30 ng/mL) and ionomycin (1 µM, Sigma-Aldrich) for 4–6 h. Expression of Ikaros and CXCR5 by targeted Jurkat lines was also assessed by flow cytometry using anti-human APC-CXCR5 (Biolegend) and anti-human PE-Ikaros (BD Biosciences). Fixation, permeabilization and intracellular staining for Ikaros and BCL-6 was performed using the Transcription Factor Buffer Set (BD Pharmingen). Cells were analyzed on a CytoFLEX flow cytometer (Beckman Coulter).

**Lentiviral shRNA-based gene targeting.** A lentiviral shRNA-based approach was employed to silence the expression of HIPK1 as well as B2M as a positive control. The lenti-shRNA vectors pGFP-C-shRNA-Lenti-Hipk1, pGFP-C-shRNA-Lenti-B2M and pGFP-C-scrambled were purchased from Origene. The packaging vectors PmD2G and PsPAX.2 were obtained from Addgene. Exponentially growing 293 T cells were split and seeded at $8 \times 10^6$ cells in 100 mm dishes in RPMI 1640 medium at 37C. The following day, cells were transfected in antibiotic- and serum-free medium with lenti shRNA plus packaging vector DNA prepared in a complex with Lipofectamine 2000. After 6 h of transfection, medium was replaced with complete serum containing RPMI medium and cells were cultured at 37C for 2 days. Human primary CD4+ T cells from healthy donors were obtained from the University of Pennsylvania Human Immunology Core and stimulated overnight with human anti-CD3- and anti-CD28-coated microbeads. Cells were harvested, de-beaded, washed with warm RPMI medium, and aliquots of $10^6$ activated CD4+ T cells were infected with 1 ml of viral supernatant collected from lenti-shRNA transfected 293T cell cultures. Polybrene was added to the viral supernatant at 8 µg/ml, cells were spin-fected at $1360 \times g$ for 1.5 h, cultured at 37C for 6 h, and restimulated with anti-CD3 and anti-CD28 beads, Activin A (100 ng/ml), IL-12 (5 ng/ml), and anti-IL-2 (2 µg/ml) to induce in vitro TFH differentiation. After 4 days of differentiation, transduced cells were FACS-sorted based on GFP expression, and expression of B2M, BCL-6, CXCR5, and PD-1 was measured by flow cytometry. In addition, sorted GFP+ in vitro TFH cells were restimulated with plate-bound human anti-CD3 and anti-CD28 (1 µg/ml each) in flat bottom 96 well plates, and supernatants were collected at the indicated timepoints for assessment of IL-21 secretion by ELISA. RNA was extracted from sorted GFP+ TFH cells using RNeasy micro kits (Qiagen), treated with DNase, and 500 ng of total RNA was reverse-transcribed using iScript cDNA synthesis kit (Bio-Rad). qRT-PCR quantification of HIPK-1, B2M and 18s rRNA transcripts was performed using Amplitaq Gold SYBR Master mix (ABI) on Applied Biosytems step one plus real- time thermo-cycler. Specific mRNA levels were determined as ratio of total 18 s rRNA. The following primer sequences were used for qRT-PCR: HIPK-1-F: CAGTCAGGA GTTCTCACGCA, HIPK-1-R: TGGCTACTTGAGGGTGGAGA, B2M-F: GCCGTGTGAACCATGTGACT, B2M-R: CATCCAATCCAAATGCGGCA, hu 18S-F: CCTTTAACGAGGATCCATTGGA, hu 18S-R: CGCTATTGGAGCT GGAATTACC.

**Pharmacologic inhibitors.** The HIPK kinase family inhibitor A64 trifluoroacetate was purchased from Sigma, and the MAP4K2 inhibitor PF06260933, which also inhibits MINK1 and TNIK, was purchased from TOCRIS. Human primary CD4+ T cells were cultured under TFH condition for 5 days in the presence of the indicated concentrations of each inhibitor (150 nM to 2500 nM for A64, 3.7 nM to 100 nM for PF06260933). In addition, anti-CD3- and anti-CD28-stimulated human CD4+ T cells (non-TFH) were cultured in the presence of inhibitors. After 5 days of primary culture, cells were harvested and $10^6$ cells were restimulated with plate-bound human anti-CD3+ anti-CD28 (1 µg/ml each) in the presence of inhibitors. Culture supernatants were collected at the indicated timepoints for measurement of IL-2 and IL-21 by ELISA (eBioscience), or RNA was extracted and analyzed for expression of the genes listed below by qRT-PCR with each primer set. Levels of each transcript were normalized to 18S RNA. Genes/primers: hABHD6 Fw: 5′-CCACGGATTCTCTGCCCACA-3′, hABHD6 Rev: 5′-TCCACGCAGAC CAAGTGCAG-3′, hARID5B Fw: 5′-GCGACCATGGCGAGGATGAA-3′, hARID 5B Rev: 5′-GTCCAGCGTGGAAGCCACAT-3′, hBCL6 Fw: 5′-CCCAACCAAGCT GAGTGCCA-3′, hBCL6 Rev: 5′-AGAGCCCGTCATGGACCTGT-3′, hBLK Fw: 5′-ATGTCGGCGCAGATTGCTGA-3′, hBLK Rev: 5′-TGCAGCACAAGGCCT CAGAC-3′, hDHRS7 Fw: 5′-ACGGCGACCTGACGCTACTA-3′, hDHRS7 Rev: 5′-GCTCCAGTCACCCACACCAC-3′, hFAS Fw: 5′-AGCCCTGTCCTCCAGGT GAA-3′, hFAS Rev: 5′-TGGGCTTTGTCTGTGTACTCCT-3′, hIKZF2 Fw: 5′-GAA GGGACGCCCTCACAGGA-3′, hIKZF2 Rev: 5′-TGCGCTGCTTGTAGCTTCGT-3′, hIKZF3 Fw: 5′-CCGCATGATGGACCAAGCCA-3′, hIKZF3 Rev: 5′-CAGG CGGTGTCTGGACCAAG-3′, hIL12RB2 Fw: 5′-CCGGAAATTGGGCTGTGGCT-3′, hIL12RB2 Rev: 5′-CAGCAACCCTGCCTCACACA-3′, hIL2RB Fw: 5′-TGGA TCTGCCTGGAGACGCT-3′, hIL2RB Rev: 5′-AGGTCGTGAACTCGCCTTGC-3′, hIL6R Fw: 5′-CGACAAGCCTCCCAGTGCAA-3′ hIL6R Rev: 5′-TGGCAATGCA GAGGAGGCGTT-3′, hPRDM1 Fw: 5′-TGCCAGGTCTGCCCACAAGAG-3′, hPR DM1 Rev: 5′-GAACTTGGCAGGGCACACCT-3′, hPTPN11 Fw: 5′-CCTCTCCC GCCTTGTACTCCA-3′, hPTPN11 Rev: 5′-TGTTGCATCAGGCCCACGTT-3′, hPTPN13 Fw: 5′-CCCTGGAGGACCAGCTGACT-3′ hPTPN13 Rev: 5′-TGCAG CATGGTGGCTGACTC-3′, hPTPN22 Fw: 5′-AGCTCATCTGGGATGTACGT GT-3′, hPTPN22 Rev: 5′-CACCAGTCCTTCCACAGCCA-3′, hSPRED2 Fw: 5′-TC CTGTGAGCACCGGAGGAT-3′.

hSPRED2 Rev: 5′-GGCATCGGCTGATCGAGGTG-3′, hTNIP1 Fw: 5′- GCCA GCTCTTCACCCACCTG-3′, hTNIP1 Rev: 5′-GGAGGTGCGCTGCTCATTCT-3′, hOAS1 Fw: 5′-AACCCAGGCCTGTGATCCTG-3′, hOAS1 Rev: 5′-CCAGGCCTC AGCCTCTTGTG-3′, hPDCD1 Fw: 5′-CTGCTCGTGGTGACCGAAGG-3′.

hPDCD1 Rev: 5′-GGGCTCATGCGGTACCAGTT-3′, hSH2D1A Fw: 5′-GCT ATTTGCTGAGGGACAGC-3′, hSH2D1A Rev: 5′-CGTGATACAGCACACATA GGC-3′, hSLAMF6 Fw: 5′-TGCTCTGTGGAGGATGCAGA-3′, hSLAMF6 Rev: 5′- GGTCCCAGGAGGACGTGAGGT-3′, hBACH2 Fw: 5′-TTGCCCTGTCCTCA GACCCA-3′, hBACH2 Rev: 5′-TGGGAGGCCGGCAATGTTCTG-3′, hBACH1 Fw: 5′-GGACACTCCTTGCCAAATGCAG-3′, hBACH1 Rev: 5′-TGACCTGGTTCTG GGCTCTCAC-3′, hIl21 Fw: 5′-AGGTCAAGATCGCCACATGA-3′, hIL21 Rev: 5′- TCAGGGACCAAGTCATTCAC-3′, hS100A8 Fw: 5′-TGAACTCTATCATCG ACGTCT-3′, hS100A8 Rev:5′-TCTAGCAATTCTTCAGGTCAT-3′, hTox2 Fw: 5′-ATGTGGGACAGCCTGGGAGA-3′, hTox2 Rev: 5′-GCTGCCAGGGCCTT CAGATA-3′, hu 18S-Fw: 5′-CCTTTAACGAGGATCCATTGGA-3′, hu 18S-Rev: 5′-CGCTATTGGAGCTGGAATTACC-3′.

**Reporting summary.** Further information on research design is available in the Nature Research Reporting Summary linked to this article.

## Data availability

Our data are available from ArrayExpress (https://www.ebi.ac.uk/arrayexpress/) with accession numbers E-MTAB-6621 (promoter-Capture-C), E-MTAB-6617 (ATAC-seq), and E-MTAB-6637 (microarray gene expression,) respectively. The source data underlying Figs. Supplementary Fig. 4, 9, and Supplementary Fig. 10 are provided as a Source Data file. Source data are provided with this paper.

## Code availability

Publicly available analysis software and code were used as described in the methods section. Source data are provided with this paper.

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

## Acknowledgements

This work was supported by The Center for Spatial and Functional Genomics at The Children's Hospital of Philadelphia (ADW and SFG), the Daniel B. Burke Endowed Chair for Diabetes Research (S.F.G.), the Jeffrey Modell Foundation (N.R.), and NIH grants AI123539 (A.D.W.), DK122586 (A.D.W. and S.F.G.), HG010067 (S.F.G.), AI146026 (N.R.), and AI115001 (N.R.).

## Author contributions

C.S. assisted with preparation of the paper and conducted the epigenomic and transcriptomic analyses with the help of E.M.; A.C. designed the custom Capture-C promoter probe set; C.L.C. provided human tonsillar T cells, N.R. contributed to the design of the immunologic studies and provided microarray data; M.E.J., M.E.L., S.L., K.M.H., and J.P. generated and sequenced the epigenomic libraries; A.T., R.M.T., P.M., and P.S. performed the CRISPR/CAS, shRNA, and pharmacologic targeting experiments; S.F.G. directed the genomic and epigenomic studies; A.D.W. directed the epigenomic and immunologic studies and wrote the manuscript. C.S., M.E.J., A.T., and R.M.T contributed equally, and S.F.G. and A.D.W. are co-senior authors.

## Competing interests

The authors declare no competing interests.
