## [Peer Review File · Nature Communications]

Reviewers' comments:

Reviewer #1 (Remarks to the Author):

Su/Johnson/Torres/Thomas et al. present the identification of open chromatin regions and promoter chromosomal contacts in TFH cells that enabled them to link SLE-associated SNPs with their putative target genes in these cells. This approach has become quite standard in the last couple of years, but it still needs to be performed in each cell type individually to extract important insights. I therefore have no reservations about the approach per se, but I do have some technical comments.

Major points:

1. I would like to see stronger evidence that TFH is indeed the causal cell type for the SLE SNPs, since their promoter connections are probed in TFH cells. Analyses such as a relative enrichment of SLE SNPs at OCRs and promoter-interacting regions across multiple cell types would be informative for this purpose. If this evidence isn't forthcoming, then the focus on SLE becomes less explicable, and perhaps the authors could integrate a wider array of GWAS traits with their TFH Capture-C data.
2. I am concerned with the very low r^2 (>0.4) used for defining proxy variants, since with such a low cutoff simply filtering these variants to fall into OCRs may not be stringent enough to filter irrelevant variants. Could the authors provide some evidence supporting their choice of r^2 ? What happens if they defer to the more commonly used 0.8? If a lot of signal is lost this way, I wonder whether the authors could make use of unthresholded summary statistics for a more robust prioritisation of target genes using approaches such as that taken in Javierre et al (ref. 17)?
3. The authors only show Chicago-processed data, but it would be good to see read-count-level bait plots. This is particularly important for fragment-level DpnII data, where coverage is likely very low, and so the question is whether Chicago calls are robust enough. I am also wondering what the total number of valid reads is for each sample.
4. The CRISPR experiments in Figure 8AB suggest that deleting the regulatory regions containing SLE-associated SNPs in Jurkat T cells leads to derepression of the genes connected to these regions in TFH. Could the authors elaborate on the potential mechanisms through which they believe these elements are acting, and whether the connections of these elements in TFH are therefore relevant for this? Perhaps they could provide some evidence (at least inferential) as to what TFs are recruited to these regions in Jurkat and TFH cells to support their model?
5. I would like to see more evidence that the functional effects of perturbing the genes in Figure 9 are relevant for SLE. Otherwise the question is what we have learned in addition to the fact that perturbing genes that are upregulated in TFH has an effect on TFH. Arriving at this conclusion does not require generating any primary data beyond gene expression. I appreciate that going beyond this is technically challenging, but at least some hints in this direction of SLE would be helpful.

Minor points:

1. The authors incorrectly claim (lines 185-188) that the difference between their method and Promoter Capture Hi-C is the use of a four-cutter enzyme. In fact, the difference between Capture-C and Capture Hi-C is in the method used to produce libraries for capture: 3C in Capture-C and Hi-C in Capture Hi-C. Both methods can and have been used with four-cutter enzymes. The main difference is that Hi-C has a biotin pulldown step that enriches for re-ligation products and should in theory lead to a much higher proportion of HiCUP's "valid reads". While this should not affect the quality of the final result (only the proportion of wasted sequencing reads), it would be useful for the community if the authors provided a summary of their HiCUP QC to facilitate a comparison

between these two methods.

2. Using binned DpnII data for Chicago is a good idea, and will probably become standard as more labs start using four-cutter-based capture approaches. For the unbinned data though, the authors mention that they use Chicago with default settings. However, for four-cutters the default minFragLen value of 150 will likely remove a few potentially useful fragments.

3. Statement at lines 162-165 is confirmatory in nature, and should be presented as such, ideally citing previous literature on this topic.

Reviewer #2 (Remarks to the Author):

This is a novel study that explores the connectome of human TFH cell promoters relative to SNPs that have been associated with lupus. As stated in the paper, the causal variants or even the causal gene have not been identified for the large majority of these SNPs, which are located in intergenic regions. The study uses two approaches:

- one identified the open chromatin regions in naïve vs. TFh CD4+ T cells, which found that 20% of the SLE SNPs are located in TFH OCRS
- the other mapped connectivity between SNPs and promoters of genes expressed in Tfh cells. Remarkably, the majority of these SNPs were connected to distal genes. Canonical genes were used to validate the results. Two genes with unknown function in Tfh cells, HIPK1 and MINK1, which were identified with this approach, were shown to control IL-21 production.

These results open a novel and rich venue to reanalyze GWAS results in SLE relative to the function of TFH cells, an important cell subset for antibody mediated diseases. It can also serve as a blue print for the analysis of other immune cell types relative to SNP datasets.

Minor edits:

define "proxy" and "sentinel" SNPs

Fig. 2C: The 3 pathways shown do not correspond exactly to what it reported in the text.

Sup. Fig 8 should be clarified showing where the deletions are, matching the SNPs and OCR1 / 2 shown on the gels

Reviewer #3 (Remarks to the Author):

The authors present a study of 3D chromosomal architecture in human tonsillar follicular helper T cells with a particular focus on interpreting SLE GWAS variants. Interpreting non-coding variants associated with disease through GWAS remains a challenge and the integrative analyses presented are interesting approaches for furthering our understanding of SLE disease biology.

There are a number of areas which I have highlighted below where additional information would make it easier for the reader to interpret the results and figures.

Major comments

1. Please include a few sentences in the discussion on the limitations of the study for example the differential gene expression analysis has been conducted on very small numbers.
2. It is challenging to interpret Figure 4.
 - a. Is the y axis on figure 4a displaying the number of "promoter-interacting region overlaps"?
 - b. Page 13 lines 244 to 250: nonprOCR are defined as "promoter-interacting OCR located in

intergenic and/intronic space". It is not clear how these can be used to conclude that "OCR not connected to a promoter" are biochemically distinct from promoter-connected OCR. From this description the nonprOCR are still interacting with a promoter. Is this referring to OCR that are not in a promoter rather than not interacting with a promoter?

c. It is not clear how the results indicate that "this promoter-Capture-C approach enriches for genomic elements that are actively engaged in gene regulation"? Enhancers are still regulating expression.

d. Why are only 14 chromatin states displayed (15 stated in legend) in Fig 4c?

e. What does genome % stand for in Fig 4c?

f. What are the red boxes in Fig 4c?

g. Please include a scale for the blue colour intensity and fold enrichment in the figure

3. On page 17, are the overlaps between the eQTL studies in SLE and the promoter-Capture-C approach more than would be expected by chance?

Minor comments

1. Clarifying the differential expression analysis

a. It would be helpful to include the number of genes that are differentially expressed between naïve and TFH cells in the text

b. The labels in figure 1B are not easy to interpret. For example does "tfh_upreg.grp" correspond to upregulated OCRs in TFH? What does "na_pos" represent? Additional details of this analysis in the figure legend would help readers unfamiliar with GSEA enrichment analysis to interpret this result.

2. Please add a key for the fold change colour gradient in figure 2c. Should this be interpreted that all the genes are upregulated in TFH compared to naïve T cells?

3. Is Figure 3A labelled correctly or should the 4 blue boxes on the top track be labelled as 4-frag instead of 1-frag?

4. Typo in figure 6 legend: "interact exclusively with on or more" should be "one or more"

5. Please include a scale for the colour gradients in Figure 7b and c. What do the orange nodes represent in figure 7c?

6. Should the y axis on the lower bar graphs in fig 8c be labelled as "percent positive cells" rather than frequency?

7. What is Tstim in figure 9g? Only TFH are mentioned in the legend.

8. In Supplementary figure 10, the OCR tracks are not very visible with the promoter-Capture-C track overlaid.

We sincerely thank the reviewers for their time, effort, and thoughtful suggestions toward improving our studies and our manuscript. We found their critiques to be particularly fair and helpful. We have incorporated the referees' input into a revision that we feel is highly valuable to the *Nature* readership across the fields of immunology, autoimmune disease, chromatin biology, and genomics. We address each comment in detail below:

Reviewer #1 (Remarks to the Author):

Su/Johnson/Torres/Thomas et al. present the identification of open chromatin regions and promoter chromosomal contacts in TFH cells that enabled them to link SLE-associated SNPs with their putative target genes in these cells. This approach has become quite standard in the last couple of years, but it still needs to be performed in each cell type individually to extract important insights. I therefore have no reservations about the approach per se, but I do have some technical comments.

We are enthused that the referee supports the approach we employed.

1. I would like to see stronger evidence that TFH is indeed the causal cell type for the SLE SNPs, since their promoter connections are probed in TFH cells. Analyses such as a relative enrichment of SLE SNPs at OCRs and promoter-interacting regions across multiple cell types would be informative for this purpose. If this evidence isn't forthcoming, then the focus on SLE becomes less explicable, and perhaps the authors could integrate a wider array of GWAS traits with their TFH Capture-C data.

The reviewer raises a very important point that we failed to make clear in our initial version of the manuscript. Indeed, we have performed LD score enrichment analysis of accessible, promoter-connected SLE SNPs across several differentiated immune cell types (GCB, TFH, Th1, Th2, Th17, Treg, monocytes, and LPS-stimulated monocytes). This analysis showed that GCB and TFH are the cell types with the highest enrichment of SLE SNPs (enrichment score of 18.9, $p=0.002$). These data confirm hundreds of immunologic studies showing that TFH are a highly relevant cell type in SLE, and show that the TFH epigenome represents a rich landscape for SLE functional genomic studies. We are incorporating this analysis into a separate manuscript that compares the 3D gene regulatory architectures of the above immune cell types across several autoimmune diseases, and therefore would prefer not to publish this entire dataset here, given it is outside the scope of the current manuscript.

However, we have now included a similar LD score comparison of SLE SNP enrichment at all OCR or all promoter-connected OCR (iOCR) in naïve CD4+ T cells vs. TFH in this revised manuscript. The results, which are presented in the results section (lines 113-118), the methods section (lines 650-663) and in a new **Supplemental Table 4**, show that SLE SNPs are enriched in the open chromatin landscape of TFH cells (15.5) compared to that of naïve CD4+ T cells (14.0). The enrichment is even greater when comparing promoter-connected OCR in TFH cells (20.3) vs. naïve cells (15.4, see lines 245-246). These data support the choice of TFH cells for our 3D variant-to-gene mapping approach used in this study, and we thank the reviewer for this suggestion.

disease_atacSet	Prop_SNPs	Prop_h2	Prop_h2_std	Enrichment	Enrich_std	Enrich_p
SLE_TFH_atac	0.017535	0.271554	0.095119	15.486	5.4244	0.003505
SLE_naive_atac	0.017643	0.247419	0.102033	14.023	5.7830	0.011160
SLE_TFH_iOCR	0.007115	0.144754	0.071080	20.343	9.9895	0.013687
SLE_naive_iOCR	0.006899	0.106275	0.066712	15.403	9.6692	0.066323

2. I am concerned with the very low r^2 (>0.4) used for defining proxy variants, since with such a low cutoff simply filtering these variants to fall into OCRs may not be stringent enough to filter irrelevant variants. Could the authors provide some evidence supporting their choice of r^2 ? What happens if they

defer to the more commonly used 0.8? If a lot of signal is lost this way, I wonder whether the authors could make use of unthresholded summary statistics for a more robust prioritisation of target genes using approaches such as that taken in Javierre et al (ref. 17)?

The reviewer raises the significant but complex issue, which we now address in more detail in the revised version of the manuscript, of how best to prioritize SNPs as a function of their linkage disequilibrium (r^2) with the sentinel SLE GWAS SNPs used in this study. We now provide all r^2 information for the reader to make an informed assessment of our findings, but we would like to make the case to retain our $r^2 > 0.4$ observations. If we restrict our analyses to proxies in tight LD ($r^2 > 0.8$) with SLE sentinel SNPs, we detect a total of 190 proxies located in 109 OCR in tonsillar T cells, and 112 of these open proxies are connected to 193 target genes. Relaxing the LD threshold to $r^2 > 0.4$ results in detection of 432 accessible proxies in 246 OCR, and 256 open SLE variants connected to 330 target genes. Therefore, including proxies in lower LD with SLE sentinels ~doubles the number of OCRs under investigation, and only increases the number of implicated target genes by 1.7-fold when we add the further constraint of insisting a promoter contact an OCR. There are several reasons, outlined below, that we think the additional proxies in lower LD are still informative for this study.

For example, **1**) in the revised **Figure 2B**, we show that genes with accessible $r^2 > 0.8$ proxies in their promoters are expressed more highly than a random sample of genes with open promoters ($P \sim 4 \times 10^{-6}$). This same analysis using an $r^2 > 0.4$ threshold still shows strong enrichment for highly expressed genes, and indeed with even higher statistical significance ($P \sim 5 \times 10^{-12}$). **2**) Pathway analysis of genes found connected to SLE proxies gives very similar results whether performed with > 0.4 or > 0.8 r^2 thresholds, and the top disease networks enriched in both cases are systemic autoimmune disorders, rheumatic disease, and type 1 diabetes. **3**) In the revised manuscript, we have addressed the statistical significance of the overlap between our physical SNP-gene associations measured in TFH cells and the statistical SNP-gene associations from prior eQTL studies using an empirical distribution hypothesis testing approach (**Supplemental Figure 8B**). To do this, we identified genes within 5Mbp from each sentinel from the most current set and used these to generate a pool of randomly-associated gene-variant pairs. We then randomly picked the same number of pairs from this random pool as we observed using our capture-C method, and combined these random pairs with the eQTL data to calculate the frequency of random overlapping gene-sentinel pairs. These steps were repeated 100,000 times to calculate an empirical distribution. We then compared the observed number of SNP-gene pair overlaps to the empirical distribution to generate a p-value. If we use an $r^2 > 0.8$ LD threshold, the expected number of overlapping sentinel-gene pairs between our Capture-C data and the eQTL studies is 1. The observed overlap in this case is 7, indicating an enrichment of 7-fold over random, with $p \sim 10^{-5}$. Using a relaxed $r^2 > 0.4$ LD threshold, the expected number of overlapping sentinel-gene pairs is 1.8, and the observed is 19, indicating a 10-fold enrichment with a p value approaching zero.

In each of the three examples above, if lowering the LD threshold were merely bringing more noise (i.e., irrelevant SNPs) into the system, we would expect to see the effect sizes decrease. Because lowering the threshold increases N, statistical significance may or may not decrease. However, we find that lowering the LD threshold increases, not decreases, the effect size and significance of the associations tested. These results indicate that including SNPs in lower LD with SLE sentinels in this case actually increases the signal-to-noise ratio of the analyses reported, presumably because it includes plenty of relevant SNPs.

Finally, **4**) several of the SNP-gene pairs that we chose for functional follow up are in very high LD with their SLE sentinel. For example, the accessible proxy rs34631447 that is connected to *BCL6* is in absolute LD ($r^2 = 1.0$) with the rs6762714 SLE sentinel, and the rs11552449 proxy connected to *HIPK1* is in very tight LD ($r^2 = 0.97$) with the rs11102701 SLE sentinel. Each of these pairs were shown to have roles in gene expression and TFH function using CRISPR/CAS9, shRNA, and/or pharmacologic inhibition. However, we also showed using CRISPR/CAS9 that the proxy rs527619 connected to the *CXCR5* promoter is indeed involved in regulation of *CXCR5* expression, even though this proxy is linked to its sentinel rs4639966 with an r^2 of 0.51. Similarly, we were able to confirm that the kinase encoded by *MINK1* is involved in TFH function, even though the proxies to which it is connected (rs71368520 and rs71368521) are linked to their sentinel (rs2286672) with an r^2 of 0.5. These results suggest that,

when combined with open chromatin-promoter connectome mapping data, proxy SNPs in LD with sentinel SNPs at a level less than $r^2=0.8$ can identify functional regulatory elements and target genes. We have included comparisons of results obtained with both LD thresholds in the revised manuscript, we have listed the r^2 values of all featured proxies in the figure legends, and have included a brief discussion of this important issue in the Methods section (lines 613-648). Again, we thank the reviewer for this important suggestion.

3. The authors only show Chicago-processed data, but it would be good to see read-count-level bait plots. This is particularly important for fragment-level DpnII data, where coverage is likely very low, and so the question is whether Chicago calls are robust enough. I am also wondering what the total number of valid reads is for each sample.

We have included new read-out-level plots for promoter Capture C at *CD28*, *CTLA4*, *ICOS*, *PTPRC*, *IKZF3*, *ERBB2*, *PGAP3*, *BCL6*, *CXCR5*, *HIPK1*, and *LSM2* in a new **Supplemental Figure 3**. The pre-processing summary of the number of valid reads for each sample is shown in the table below. The average of total number of valid reads per library is 600 million. We have included this information in **Supplemental Figure 3**.

condition	sample	input	paired_align	paired_align	valid_pair	valid_pair	ri	same_circul	same_dangli	same_interni	reiligation	ri	contiguous	s	wrong_size	unique_ditag	unique_ditag	captured	captured	rat_cis	captured	cis	captured	trans	captun
TFH	TFH_rep1	2939034738	1644775526	55.96	602163591	36.61	0.65	6.19	31.75	24.79	0	0	111954790	18.59	85863749	76.7	54415395	63.37	36.63						
TFH	TFH_rep2	2972871343	1658287890	55.78	530510248	31.99	0.31	5.48	37.58	24.64	0	0	152419541	28.73	107385810	70.45	73463930	68.41	31.59						
TFH	TFH_rep3	2753741693	1488119213	53.86	692603994	46.7	1.12	4.92	37.54	9.72	0	0	247128786	35.68	172472728	69.79	92992391	53.92	46.08						
naiveT	Naive_rep1	2969804255	1652422445	55.64	638560154	38.64	0.63	5.24	39.3	16.18	0	0	258299705	40.45	183842739	71.17	104841378	57.03	42.97						
naiveT	Naive_rep2	2916678759	1611852080	55.26	516981205	32.07	0.26	4.85	40.73	22.09	0	0	91888448	17.77	59113148	64.33	40595148	68.67	31.33						
naiveT	Naive_rep3	2977521225	1656989896	55.65	623661783	37.64	3.51	5.32	43.11	10.43	0	0	263596939	42.27	185800896	70.49	91543813	49.27	50.73						

4. The CRISPR experiments in Figure 8AB suggest that deleting the regulatory regions containing SLE-associated SNPs in Jurkat T cells leads to derepression of the genes connected to these regions in TFH. Could the authors elaborate on the potential mechanisms through which they believe these elements are acting, and whether the connections of these elements in TFH are therefore relevant for this? Perhaps they could provide some evidence (at least inferential) as to what TFs are recruited to these regions in Jurkat and TFH cells to support their model?

We addressed this issue in the discussion (lines 394-414): “Importantly, we were able to validate direct roles for several SLE-associated distal OCR in the regulation of their connected genes (*BCL6*, *CXCR5*, *IKZF1*) using CRISPR/CAS9-mediated editing in human T cells, suggesting that SLE-associated genetic variation marks regulatory elements for genes with known roles in TFH and/or SLE biology. A locus control region ~130 kb upstream of the *BCL6* gene has been defined previously in germinal center B cells³⁶, and we also find evidence for usage of this region by human TFH cells at the level of open chromatin, histone enhancer marks, and long-range connectivity to the *BCL6* promoter (**Supplemental Figure 10**). However, we also observe a much more distant ‘stretch’ enhancer within the *LPP* gene in TFH cells, as evidenced by extensive open chromatin, H3K27 acetylation, and H3K4 mono-methylation (**Supplemental Figure 10**). This region shows extensive connectivity with *BCL6* in the 3D architecture of the nucleus, and the 1 Mb distal SLE-associated *BCL6* enhancer validated by genome editing in this study is contained within this *BCL6* stretch enhancer. This enhancer region is occupied in lymphoid cell lines by NFkB/RelA and POU2F2, both transcription factors known to positively regulate immunoglobulin and inflammatory gene expression (ENCODE). Long-range regulatory elements for *CXCR5* have not been previously identified, and the -180 kb SLE-associated element in this study is the first validated for *CXCR5*. Deletion of this element led to increased expression of *CXCR5*, suggesting that unlike the distal *BCL6-LPP* enhancer, this element is a silencer in Jurkat cells. Consistent with this finding, this region is occupied by the repressive transcription factors YY1, BHLHE40 and BATF in lymphoid cells (ENCODE), but its function in primary TFH cells remains to be determined.”

We suspect that these regulatory regions may have different functions in TFH vs. Jurkat cells, and we plan to target these regions specifically in TFH cells and other T helper subtypes in future studies.

5. I would like to see more evidence that the functional effects of perturbing the genes in Figure 9 are relevant for SLE. Otherwise the question is what we have learned in addition to the fact that perturbing genes that are upregulated in TFH has an effect on TFH. Arriving at this conclusion does not require generating any primary data beyond gene expression. I appreciate that going beyond this is technically challenging, but at least some hints in this direction of SLE would be helpful.

The referee raises an important question as to the relevance of HIPK1- and MINK1-regulated pathways to SLE. We show that both HIPK1 and MINK1 regulate IL-21 production by TFH cells, but we did not go into detail about the significance of this to SLE. IL-21 is absolutely required for lupus in mouse models, and SLE patients have elevated levels of IL-21 produced by a pathogenic subset of TFH that drives plasma cell differentiation (Wang Nat. Comm. 2018, Dolff Arthritis Res. & Therapy 2011). We have clarified this in the revised manuscript.

However, we agree that more information on how HIPK1 and MINK1 impact the expression of additional genes linked to SLE would improve the significance of our findings. Therefore, we have performed additional studies to measure the impact of HIPK and MINK inhibition on the expression of over 20 genes linked to SLE and/or TFH biology (Tsokos, Nat. Rev. Rheum. 2016) by human TFH cells using quantitative PCR (*PTPN22*, *PTPN11*, *PTPN13*, *FAS*, *PDCD1*, *IL-6R*, *SPRED2*, *DHRS7*, *PRDM1*, *TNIP1*, *ABHD6*, *ARID5B*, *IL2RB*, *BACH2*, *IKZF2*, *IKZF3*, *BLK*, *IL12RB2*, *OAS1*, *SH2D1A*, *SLAMF6*, *TOX2*, and *S100A8*). We now show that at least 7 of these SLE-associated genes in addition to *IL21* are regulated by HIPK1 in human TFH cells. These experiments are described in the methods (lines 789-828) data are presented in the results section (lines 363-369) and in a new panel G in **Figure 9**.

Indeed, we plan to follow up these studies with additional genetic and pharmacologic targeting of these kinases in T helper subtypes in future studies, and we are in the process of generating a conditional HIPK1 knockout mouse (which does not currently exist). We thank the referee for this helpful suggestion.

6. The authors incorrectly claim (lines 185-188) that the difference between their method and Promoter Capture Hi-C is the use of a four-cutter enzyme. In fact, the difference between Capture-C and Capture Hi-C is in the method used to produce libraries for capture: 3C in Capture-C and Hi-C in Capture Hi-C. Both methods can and have been used with four-cutter enzymes. The main difference is that Hi-C has a biotin pulldown step that enriches for re-ligation products and should in theory lead to a much higher proportion of HiCUP's "valid reads". While this should not affect the quality of the final result (only the proportion of wasted sequencing reads), it would be useful for the community if the authors provided a summary of their HiCUP QC to facilitate a comparison between these two methods.

We have corrected this error in the text (lines 150-157). We have reported a HiCUP summary of valid reads in a new **Supplemental Table 8**.

7. Using binned DpnII data for Chicago is a good idea, and will probably become standard as more labs start using four-cutter-based capture approaches. For the unbinned data though, the authors mention that they use Chicago with default settings. However, for four-cutters the default minFragLen value of 150 will likely remove a few potentially useful fragments.

We used unbinned DpnII data for interaction calling, but the background noise estimation used a binsize setting of 2500 for 1frag, and a binsize setting of 10000 for 4frag. As the reviewer points out, the minFragLen value is default value of 150 for both 1frag and 4frag. Although we did indeed miss some interactions due to high minFragLen in the 1frag analysis, we rescued these in our 4frag resolution analysis.

8. Statement at lines 162-165 is confirmatory in nature, and should be presented as such, ideally citing previous literature on this topic.

Agreed. We did cite other seminal studies that have used ATAC-seq/DNase-seq/ChIP-seq to identify putatively functional variants (Javierre et al, Onengut-Gumuscu et al) in the original version of the manuscript, but should have explicitly referred them in this statement. We have done so, and the opening of this section now reads “These results confirm...” (lines 136-139).

Reviewer #2 (Remarks to the Author):

This is a novel study that explores the connectome of human TFH cell promoters relative to SNPs that have been associated with lupus. As stated in the paper, the causal variants or even the causal gene have not been identified for the large majority of these SNPs, which are located in intergenic regions. The study uses two approaches:

- one identified the open chromatin regions in naïve vs. TFh CD4+ T cells, which found that 20% of the SLE SNPs are located in TFH OCRS
- the other mapped connectivity between SNPs and promoters of genes expressed in Tfh cells. Remarkably, the majority of these SNPs were connected to distal genes. Canonical genes were used to validate the results. Two genes with unknown function in Tfh cells, HIPK1 and MINK1, which were identified with this approach, were shown to control IL-21 production.

These results open a novel and rich venue to reanalyze GWAS results in SLE relative to the function of TFH cells, an important cell subset for antibody mediated diseases. It can also serve as a blue print for the analysis of other immune cell types relative to SNP datasets.

We appreciate the referee’s enthusiasm for our approach.

1. Define “proxy” and “sentinel” SNPs

Sentinel SNPs are SNPs that yielded the P-value for association at a given locus. These are commonly not the causal variant themselves, but rather act as a tag for the underlying causal variant(s). Proxy SNPs are those SNPs that are in linkage disequilibrium with sentinel SNPs, some of which are expected to be causal. In other words, the frequency of co-inheritance is measured as linkage disequilibrium, and is generally defined as SNPs linked with sentinel SNPs above a high (e.g. $r^2 > 0.8$) or moderate (e.g. $r^2 > 0.4$) threshold. We used a moderate threshold for these studies given we could triage most of the LD-proxy SNPs that did not obey our two constraints of insisting on both being open and being contacted by a gene promoter at high resolution. We have included definitions of these standard terms in the GWAS field in the methods section (lines 613-648), along with our rationale for the LD thresholds used.

2. Fig. 2C: The 3 pathways shown do not correspond exactly to what it reported in the text.

This error has been fixed in the text of the revised manuscript (line 134). The figure is correct.

3. Sup. Fig 8 should be clarified showing where the deletions are, matching the SNPs and OCR1/2 shown on the gels.

The deletions and SNPs are shown in the tracks in the Figure. We have enlarged some of the text and highlighted the open proxy SNPs in red in the revised version of **Supplemental Figure 9**.

Reviewer #3 (Remarks to the Author):

The authors present a study of 3D chromosomal architecture in human tonsillar follicular helper T cells

with a particular focus on interpreting SLE GWAS variants. Interpreting non-coding variants associated with disease through GWAS remains a challenge and the integrative analyses presented are interesting approaches for furthering our understanding of SLE disease biology. There are a number of areas which I have highlighted below where additional information would make it easier for the reader to interpret the results and figures.

1. Please include a few sentences in the discussion on the limitations of the study for example the differential gene expression analysis has been conducted on very small numbers.

We have included the following language in the introduction: “This approach, which we used recently to identify novel genes at bone mineral density loci¹⁴, only requires three replicate samples to make valid interaction calls, and it does not require material from SLE patients or genotyped individuals. By design, this approach does not determine the effect of a variant in the system, but rather, uses reported variants as ‘signposts’ to identify potential gene enhancers in normal tissue” (lines 64-68). We also include the following text in the discussion: “This study did not determine the impact of SLE-associated genetic variation on chromatin accessibility or promoter interactions. Rather, we use the location of reported SLE variants as ‘signposts’ to identify open chromatin elements that may regulate SLE-relevant gene expression in the context of normal follicular helper T cell biology” (lines 379-383).

2. It is challenging to interpret Figure 4.

a. Is the y axis on figure 4a displaying the number of “promoter-interacting region overlaps”?

Correct. The number of promoter-interacting region (PIRs) overlapping with each histone mark or ATAC-seq feature is depicted, as indicated in the figure and the legend.

b. Page 13 lines 244 to 250: nonprOCR are defined as “promoter-interacting OCR located in intergenic and/intronic space”. It is not clear how these can be used to conclude that “OCR not connected to a promoter” are biochemically distinct from promoter-connected OCR. From this description the nonprOCR are still interacting with a promoter. Is this referring to OCR that are not in a promoter rather than not interacting with a promoter?

This is correct. As indicated in the results section, about 30% of the OCRs detected in our study are connected to a promoter as measured by Capture-C. In this section we are comparing the set of promoter-connected intergenic/intronic OCR to the set of intergenic/intronic OCR that are not connected to a promoter. We have clarified this section in the revised manuscript (lines 205-210).

c. It is not clear how the results indicate that “this promoter-Capture-C approach enriches for genomic elements that are actively engaged in gene regulation”? Enhancers are still regulating expression.

The current model indicates that distal enhancers function to regulate gene expression by looping in space to physically interact with promoters. In **Figure 4B**, we compared the OCR that are involved in promoter interactions (iOCR) to OCR that are not. Consistent with this current model, we found that iOCR engaged in promoter interactions as measured by Capture-C show the highest levels of active histone marks, and are distinguishable from OCR not physically connected to a promoter. While ‘enhancers’ as defined in general as OCR flanked by active histone marks may or may not be actively engaged in gene regulation, our data suggest that OCR connected to promoters at the time of analyses are actively engaged in the process of regulating gene expression.

d. Why are only 14 chromatin states displayed (15 stated in legend) in Fig 4c?

Fourteen chromHMM states out of the 15 state model are depicted. One state that did not fit into a recognizable chromatin signature was not included in the figure for clarity. The legend has been corrected (lines 1010-1013), and the methods section has been clarified (lines 689-697).

e. What does genome % stand for in Fig 4c?

The relative percentage of whole genome are represented by different chromatin state. Comparing iOCR/prOCR/nonprOCR to “genome%” inform us that those OCRs are enriched at promoter- or enhancer- relevant chromatin states instead of “neutral” chromatin. Details refer to chromHMM paper (PMID: 29120462).

f. What are the red boxes in Fig 4c?

These were added manually to aid with visual comparison, but have been eliminated in order to avoid any confusion on the part of the reader.

g. Please include a scale for the blue colour intensity and fold enrichment in the figure.

The color intensity in the chromoHMM analysis has no absolute scale (PMID: 29120462), but represents relative level of enrichment for each feature. We have clarified this in the legend (lines 1010-1013) to indicate that higher color intensity means higher enrichment of the given chromatin state on the y-axis within each set shown on the x-axis.

3. On page 17, are the overlaps between the eQTL studies in SLE and the promoter-Capture-C approach more than would be expected by chance?

This is an excellent question. In the revised manuscript, we now address the statistical significance of the overlap between our physical SNP-gene associations measured in TFH cells and the statistical SNP-gene associations from prior eQTL studies. Two challenges to these comparisons should be noted: **1)** Both eQTL studies used different sentinel sets from the updated set we use in this manuscript, and **2)** all these studies use different cell types. The study by Odhams used a colocalization method on lymphoblastoid cell lines (LCLs), while the Bentham study used an RTC scoring method to identify eQTL among multiple cell types including NK cells, monocytes, monocytes stimulated with LPS or interferon, B cells, and CD4+ T cells. Our variant-gene mapping is based on physical interactions between SNPs and genes in TFH cells. Theoretically, the same variants may regulate different genes in different cell types. However, to achieve the fairest possible statistical comparisons between these groups, we used an empirical distribution hypothesis testing approach. We identified genes within 5Mbp from each sentinel from the most current set and used these to generate a pool of randomly-associated gene-variant pairs. We then randomly picked the same number of pairs from this random pool as we observed using our Capture-C method, and combined these random pairs with the eQTL data to calculate the frequency of random overlapping gene-sentinel pairs. These steps were repeated 100,000 times to calculate an empirical distribution. We then compared the observed number of SNP-gene pair overlaps to the empirical distribution to generate a p-value. If we use an $r^2 > 0.8$ LD threshold, the expected number of overlapping sentinel-gene pairs between our Capture-C data and the eQTL studies is 1. The observed overlap in this case is 7, indicating an enrichment of 7-fold over random, with $P \sim 10^{-5}$. Using a relaxed $r^2 > 0.4$ LD threshold, the expected number of overlapping sentinel-gene pairs is 1.8, and the observed is 19, indicating a 10-fold enrichment with a p value approaching zero. We have included this analysis in the results section (lines 297-298), the methods section (lines 693-711), and **Supplemental Figure 8B**. We thank the referee for this suggestion.

4. Clarifying the differential expression analysis

a. It would be helpful to include the number of genes that are differentially expressed between naïve and TFH cells in the text

There are 1,496 differentially expression genes (DEG, $P < 0.01$, $FC > 2$) between naïve and TFH. This has been included in the text of the results section.

b. The labels in figure 1B are not easy to interpret. For example, does “tfh_upreg.grp” correspond to upregulated OCRs in TFH? What does “na_pos” represent? Additional details of this analysis in the figure legend would help readers unfamiliar with GSEA enrichment analysis to interpret this result.

GSEA enrichment analysis was performed to evaluate whether the set of differentially expressed genes (tfh_upreg.grp or tfh_down.grp) between TFH and naïve by our microarray data were overrepresented at the top or bottom of a gene list pre-ranked by their promoter accessibility (log fold change) detected in our ATAC-seq data. This comparison informs us the consistent relationship between gene expression and gene promoter accessibility.

When gene set that are higher expressed in TFH (tfh_upreg.grp) were compared to promoter accessibility pre-ranked genes, gene set were enriched at the top (positive $\log FC(\text{TFH}/\text{Naïve})$ promoter accessibility change) of the ranked list. When gene set that are lower expressed in TFH (tfh_downreg.grp) were compared to pre-ranked genes by promoter accessibility, gene set were enriched at bottom (negative $\log FC(\text{TFH}/\text{Naïve})$ promoter accessibility change) of ranked list. Thus majority of gene expression change show the same direction of their promoter accessibility change.

Since we used user-defined pre-ranked gene list, the default output could not recognize what the rank is based on (in our case, it is accessibility change rank) and put “na_pos” and “na_neg” instead. In our case, “na_pos” and “na_neg” means where the differentially expressed genes are located in the accessibility change rank. “na_pos” can be replaced by “Promoter accessibility increase” while “neg_neg” can be replaced by “Promoter accessibility decrease”. The third part of the graph (bottom with gray part) shows how promoter accessibility $\log FC$ is distributed along the list. We have revised the Figure and legend (lines 982-985) to clarify these issues.

5. Please add a key for the fold change colour gradient in figure 2c. Should this be interpreted that all the genes are upregulated in TFH compared to naïve T cells?

The color scale refers to differential promoter accessibility. A scale bar has been added to the Figure, and this has been clarified in the legend (lines 993-994).

6. Is Figure 3A labelled correctly or should the 4 blue boxes on the top track be labelled as 4-frag instead of 1-frag?

The Figure is labelled correctly. The top row depicts the higher resolution one fragment scheme while the lower row depicts the lower resolution analysis in which 4 adjacent fragments are concatenated in silico into 1 larger fragment.

7. Typo in figure 6 legend: “interact exclusively with on or more” should be “one or more”

This typo has been corrected (line 1038).

8. Please include a scale for the colour gradients in Figure 7b and c. What do the orange nodes represent in figure 7c?

A scale has been added to the figure and the legend has been clarified (lines 1047-1050).

9. Should the y axis on the lower bar graphs in fig 8c be labelled as “percent positive cells” rather than frequency?

Agreed. % positive cells would clarify and we have modified the Figure as suggested.

10. What is Tstim in figure 9g? Only TFH are mentioned in the legend.

‘Tstim’ and ‘Tact’ are equivalent shorthand for naïve CD4+ T cells stimulated under neutral, non-TFH skewing conditions, which serve as a control for the TFH condition. We have made the Figure labels more consistent, and have clarified this in the legend (line 1074).

11. In Supplementary figure 10, the OCR tracks are not very visible with the promoter-Capture-C track overlaid.

We have included a modified version of **Supplemental Figure 10** in which the ATAC-seq tracks are flipped below this axis in order to be more visible.

REVIEWERS' COMMENTS:

Reviewer #1 (Remarks to the Author):

My point 1 - I thank the authors for providing these results. Admittedly, I am not a huge fan of the key motivating evidence (i.e., GWAS signal enrichment analysis across multiple cell types) being held back for a future study, but at least the enrichment in TFH over naive is shown. I leave this point to the editor's discretion.

My point 2 - Thanks for this analysis. With such a low r^2 cutoff, the chance of spurious associations clearly increases (while p-values become lower due to higher statistical power). However, it seems that some reasonable signal is still gained this way and at least the authors now report what gene was called with what cutoff. Part 3 of the response to my point (validation by eQTL-eGene relationships) is interesting, but I didn't quite get what it had to do with the point I raised - could the authors please clarify?

My point 3 - Thanks for reporting raw data plots. I'd just like to check what Chicago cutoff do blue points correspond to, because by default they correspond to that of 3, which is conventionally considered "subthreshold" and not "significant" interactions. If blue corresponds to scores ≥ 5 , it makes sense to mention this explicitly in the legend.

My point 4 - It is good that the authors have now discussed this point. I remain wondering to what extent analysis in Jurkat cells is relevant, since the authors themselves have now convinced us that there's clearly some TFH-specific biology in SLC. CRISPR methods for native blood cell perturbation are becoming feasible. I appreciate however that redoing validation in TFH is likely too much of an ask. In part, this makes me question why there's an expectation to include "validation" analyses into papers of this kind, even when they are necessarily selective and not always feasible in the correct cell type / setting. This is a theoretical point though (echoing Tim Hughes, J Biol 2009) and not a suggestion to remove these data.

My point 5 - This analysis adequately addresses my point, thanks.

My point 6 - Thanks for providing the summary statistics. I note that as expected the proportions of valid reads are very low compared with Capture Hi-C. However, as I mentioned initially this only reflects the low efficiency of Capture-C, and not the reliability of the signals that are recovered using this method.

Reviewer #3 (Remarks to the Author):

I'd like to thank the authors for their response, all my queries have been addressed.

POINT-BY-POINT RESPONSE TO REFEREE COMMENTS AND EDITORIAL REQUESTS

REVIEWERS' COMMENTS:

Reviewer #1 (Remarks to the Author):

My point 1 - I thank the authors for providing these results. Admittedly, I am not a huge fan of the key motivating evidence (i.e., GWAS signal enrichment analysis across multiple cell types) being held back for a future study, but at least the enrichment in TFH over naive is shown. I leave this point to the editor's discretion.

RESPONSE: We agree with the referee that the enrichment over naive is quite meaningful in this context, and share his/her enthusiasm for the larger comparison across more cell types. However, we disagree that GWAS signal enrichment is the key motivating evidence for choosing follicular helper T cells for our study. GWAS signal enrichment in TFH is certainly one piece of evidence, but a larger motivation is the over 2000 publications showing that follicular helper T cells are associated with and have causal roles in humoral immunity and autoantibody-mediated inflammatory disorders.

My point 2 - Thanks for this analysis. With such a low r^2 cutoff, the chance of spurious associations clearly increases (while p-values become lower due to higher statistical power). However, it seems that some reasonable signal is still gained this way and at least the authors now report what gene was called with what cutoff. Part 3 of the response to my point (validation by eQTL-eGene relationships) is interesting, but I didn't quite get what it had to do with the point I raised - could the authors please clarify?

RESPONSE: The eQTL-eGene analysis was requested by reviewer 3. Lowering the LD threshold to 0.4 increases the number of eQTL-pCapC overlaps and increases the significant enrichment score, indicating to us that valid SNPs are to be found at lower LD's without simply introducing noise.

My point 3 - Thanks for reporting raw data plots. I'd just like to check what Chicago cutoff do blue points correspond to, because by default they correspond to that of 3, which is conventionally considered "subthreshold" and not "significant" interactions. If blue corresponds to scores ≥ 5 , it makes sense to mention this explicitly in the legend.

RESPONSE: We indeed used a threshold CHICAGO score of 5, and have made this clear in the figure legends in the revised manuscript.

My point 4 - It is good that the authors have now discussed this point. I remain wondering to what extent analysis in Jurkat cells is relevant, since the authors themselves have now convinced us that there's clearly some TFH-specific biology in SLC. CRISPR methods for native blood cell perturbation are becoming feasible. I appreciate however that redoing validation in TFH is likely too much of an ask. In part, this makes me question why there's an expectation to include "validation" analyses into papers of this kind, even when they are necessarily selective and not always feasible in the correct cell type / setting. This is a theoretical point though (echoing Tim Hughes, J Biol 2009) and not a suggestion to remove these data.

My point 5 - This analysis adequately addresses my point, thanks.

My point 6 - Thanks for providing the summary statistics. I note that as expected the proportions of valid reads are very low compared with Capture Hi-C. However, as I mentioned initially this only reflects the low efficiency of Capture-C, and not the reliability of the signals that are recovered using this method.

RESPONSE: We entirely concur with the referee that no modifications are required regarding points 4 through 6.

Reviewer #3 (Remarks to the Author):

I'd like to thank the authors for their response, all my queries have been addressed.

RESPONSE: We thank the referee for his/her time and effort in helping to improve the manuscript.